# Capture Hi-C identifies putative target genes at 33 breast cancer risk loci

Joseph S. Baxter [1], Olivia C. Leavy[2,3], Nicola H. Dryden[1], Sarah Maguire[1], Nichola Johnson[1], Vita Fedele [1], Nikiana Simigdala[1], Lesley-Ann Martin[1], Simon Andrews[4], Steven W. Wingett[4], Ioannis Assiotis[5], Kerry Fenwick[5], Ritika Chauhan[1], Alistair G. Rust[1], Nick Orr[1], Frank Dudbridge[2,3], Syed Haider[1] & Olivia Fletcher [1]

Genome-wide association studies (GWAS) have identified approximately 100 breast cancer risk loci. Translating these findings into a greater understanding of the mechanisms that influence disease risk requires identification of the genes or non-coding RNAs that mediate these associations. Here, we use Capture Hi-C (CHi-C) to annotate 63 loci; we identify 110 putative target genes at 33 loci. To assess the support for these target genes in other data sources we test for associations between levels of expression and SNP genotype (eQTLs), disease-specific survival (DSS), and compare them with somatically mutated cancer genes. 22 putative target genes are eQTLs, 32 are associated with DSS and 14 are somatically mutated in breast, or other, cancers. Identifying the target genes at GWAS risk loci will lead to a greater understanding of the mechanisms that influence breast cancer risk and prognosis.

[1] Breast Cancer Now Toby Robins Research Centre, The Institute of Cancer Research, London SW3 6JB, UK. [2] Department of Non-communicable Disease Epidemiology, The London School of Hygiene and Tropical Medicine, London WC1E 7HT, UK. [3] Department of Health Sciences, University of Leicester, Leicester LE1 7RH, UK. [4] Bioinformatics Group, The Babraham Institute, Cambridge CB22 3AT, UK. [5] Tumour Profiling Unit, The Institute of Cancer Research, London SW3 6JB, UK. These authors contributed equally to this work: Joseph S. Baxter, Olivia C. Leavy, Nicola H. Dryden. These authors jointly supervised this work: Syed Haider, Olivia Fletcher. Correspondence and requests for materials should be addressed to S.H. (email: Syed.Haider@icr.ac.uk) or to O.F. (email: Olivia.Fletcher@icr.ac.uk)

Genome-wide association studies (GWAS) coupled with large-scale replication and fine-mapping studies have led to the identification of approximately 100 breast cancer risk loci. Breast cancer is a heterogeneous disease with two main subtypes defined by the presence (ER+) or absence (ER−) of the oestrogen receptor[1]. Approximately 80% of newly diagnosed breast cancers are ER+ although this proportion varies with age at diagnosis and ethnicity[1]. The majority of breast cancer GWAS risk loci have been identified on the basis of their association with overall breast cancer risk, or risk of ER+ disease[2]. Most of the risk single nucleotide polymorphisms (SNPs) map to non-protein-coding regions and are thought to influence transcriptional regulation[3,4]; many map to gene deserts with the nearest known protein-coding genes mapping several hundred kilobases (kb) away. Translating these findings into a greater understanding of the mechanisms that influence an individual woman's risk requires the identification of causal variants and the targets of these causal variants (i.e. genes or non-coding RNAs that mediate the associations observed in GWAS. Systematic approaches to the functional characterisation of cancer risk loci have been proposed[4,5]. These include fine mapping of potentially large genomic regions (defined as regions that include all SNPs correlated with the published SNP with an $r^2 \leq 0.2$), the analysis of SNP genotypes in relation to expression of nearby genes (eQTL) and the use of chromatin association methods (chromosome conformation capture (3 C) and Chromatin Interaction Analysis by Paired-End Tag Sequencing (ChIA-PET)) of regulatory regions to determine the identities of target genes. To facilitate a high-throughput approach to the identification of target genes at GWAS risk loci we developed Capture Hi-C (CHi-C)[6]. This novel Hi-C protocol[7] allows high-throughput, high-resolution analysis of physical interactions between regulatory elements and their target genes. We have used CHi-C previously, to characterise three breast cancer risk loci mapping to gene deserts at 2q35, 8q24.21 and 9q31.2[6]. Here we selected 63 established breast cancer risk loci (Supplementary Data 1); we identify CHi-C interaction peaks involving 110 putative target genes mapping to 33 loci and demonstrate long-range interaction peaks some of which span megabase (Mb) distances and involve adjacent risk loci. All CHi-C interaction peaks can be viewed at bit.ly/CHiC-BC. We carry out eQTL analyses, analyses of disease-specific survival (DSS) and compare our putative target genes with somatically mutated cancer genes to assess the orthogonal support for these putative target genes. High-throughput CHi-C analysis can contribute to on-going efforts to functionally annotate GWAS risk loci.

## Results

### Generating CHi-C libraries for 63 breast cancer risk loci. We generated CHi-C libraries in two ER+ breast cancer cell lines (T-

47D, ZR-75-1), two ER− breast cancer cell lines (BT-20, MDA-MB-231), one "normal" breast epithelial cell line (Bre80-Q-TERT (Bre80)) and a control, non-breast lymphoblastoid cell line (GM06990) (Supplementary Fig. 1). We defined an interaction peak as any pair of HindIII fragments for which the number of di-tags was significantly (FDR adjusted outlier test $P < 0.01$) greater than expected under a negative binomial model, taking into account both the distance between the HindIII fragments and the propensity of the bait fragment to form interactions ("interactability"; Methods). The number of di-tags that constituted an interaction peak depended on the distance between the interacting fragments and ranged from 5 to 14,151. We defined a locus as a single continuous capture region, annotated by at least one risk SNP (Methods).

**Distribution of CHi-C interaction peaks at the 63 risk loci**. The number of interaction peaks at each locus, in each cell line ranged from zero to 1,744 (Supplementary Data 2, Supplementary Fig. 2), with two outliers (1,744 at 8q21.11-rs2943559 in ZR-75-1 and 1,007 at 8q24.21-rs13281615 in T-47D). There were 12 loci (19.0%) at which there were no interaction peaks in any of the cell lines we examined (Supplementary Data 2); these loci were excluded from further analyses. 46 (90.2%) of the 51 loci that we were able to analyse were identified on the basis of their association with overall breast cancer risk or risk of ER+ disease; the exceptions were 2p24.1-rs12710696, 5p15.33-rs10069690, 6q25.1-rs12662670, rs2046210, 16q12.2-rs11075995 and 19p13.11-rs8170 which were associated with ER− and/or triple negative breast cancer (TNBC)[8–20].

We first tested for differences in the median number of interaction peaks across the six different cell lines according to ER status and cell type (breast/non-breast). The median number of interaction peaks per locus varied significantly between cell lines (Kruskal–Wallis test $P = 0.0006$; Table 1, Fig. 1a). There were, on average, more statistically significant interaction peaks per locus in the ER+ breast cancer cell lines (T-47D, ZR-75-1) compared to the ER− breast cancer cell lines (BT-20, MDA-MB-231, Mann–Whitney test $P = 0.0008$) or the control lymphoblastoid cell line (GM06990, Mann–Whitney test $P = 0.002$). There was, however, no difference between the number of interaction peaks per locus in the ER+ breast cancer cell lines and the normal mammary epithelial cell line (Bre80, Mann–Whitney test $P = 0.85$; Table 1), consistent with Bre80 representing a progenitor cell population that gives rise to ER+ breast cancer cells. Similarly, the median distance between interacting fragments varied across cell lines (Kruskal–Wallis test $P < 1 \times 10^{-6}$; Table 1, Fig. 1b) with a greater proportion of longest range interaction peaks (>2 Mb) in the ER+ breast cancer cell lines

### Table 1 Characteristics of 51 informative risk loci in six cell lines

| Cell line | T-47D | ZR-75-1 | Bre80[a] | BT-20 | MDA-MB-231 | GM06990[b] |
|---|---|---|---|---|---|---|
| Origin | Breast | Breast | Breast | Breast | Breast | Lymphoblastoid |
| Cancer/normal | Cancer | Cancer | Normal | Cancer | Cancer | Normal |
| Receptor status | ER+ | ER+ | ER− | ER− | ER− | ER− |
| Breast cancer subtype | Luminal | Luminal | Normal | Basal A | Basal B | N/A |
| Informative loci (%) | 34 (66.7) | 38 (74.5) | 41 (80.4) | 21 (41.2) | 27 (52.9) | 23 (45.1) |
| Median peaks per locus (range) | 7 (0–1,107) | 9 (0–1,744) | 10 (0–181) | 0 (0–246) | 1 (0–466) | 0 (0–155) |
| Median distance between interacting fragments[c] (kb) | 1392 | 1647 | 349 | 338 | 388 | 534 |
| Number (%) of peaks >2000 kb | 1453 (35.4) | 1417 (36.4) | 94 (7.3) | 128 (9.7) | 108 (8.0) | 102 (12.5) |

[a] Bre80-Q-TERT (Bre80) are normal Bre80 TERT-immortalised mammary epithelial cells, kindly provided by Prof Georgia Chenevix-Trench (Queensland Institute of Medical Research, Brisbane, Queensland, Australia)
[b] GM06990 are Epstein-Barr virus transformed B-lymphocytes from the Coriell Cell Repositories (Coriell Institute for Medical Research, New Jersey, USA)
[c] Range is not given as it was pre-defined to be 10 kb to 5 Mb

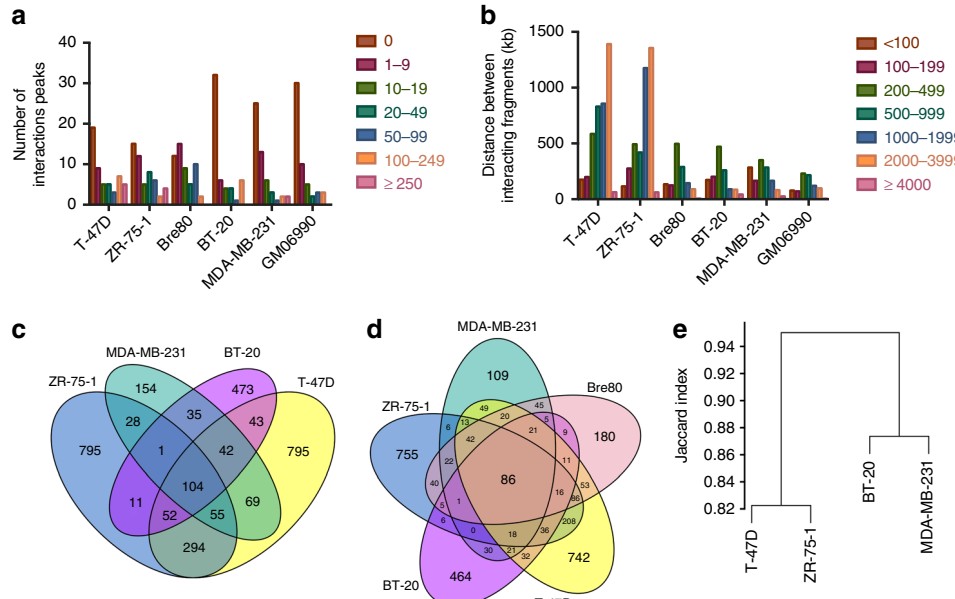

**Fig. 1** Cell-type-specificity of interaction peaks at 51 informative breast cancer risk loci. Bar charts showing (**a**) the number of loci at which there were 0, 1–9, 10–19, 20–49, 50–99, 100–249 or >250 interaction peaks and (**b**) the number of interaction peaks at which the distance between interacting fragments are <100, 100–199, 200–499, 500–999, 1,000–1,999, 2,000–3,999, ≥4,000 kb for each cell line analysed. **c**, **d** Venn diagrams illustrating the overlap between interacting fragments in (**c**) the four breast cancer cell lines and (**d**) the five breast cell lines. **e** Dendogram showing Jaccard dissimilarity scores (i.e. 1—similarity coefficient) for the four breast cancer libraries

compared to all other cell lines. In an analysis of 51 breast cancer cell lines, there was no evidence that luminal (ER+) cell lines carried more genome aberrations than basal (ER−) cell lines[21]. We cannot, however, exclude the possibility that rearrangements, gains and losses occur preferentially at ER+ risk loci in ER+ cell lines and this contributes to the higher proportion of long-range interaction peaks we observe in the T-47D and ZR-75-1 breast cancer cell lines.

To gain further insight into the relationship between interaction peaks and cell-type-specificity, we looked at the number of interaction peaks that were identical between two or more cell types. Of the 12,736 interaction peaks, we identified 7,681 (60.3%) were present in a single cell line and 5,055 (39.7%) were present in multiple cell lines (FDR adjusted outlier test $P < 0.01$ Supplementary Table 1). The subset of interaction peaks that were present in at least two cell lines are provided as supplementary data (Supplementary Data 3). Excluding two loci with large outliers (i.e. 8q21.11-rs2943559 and 8q24.21-rs13281615 at which there were 1,744 and 1,007 interaction peaks in T-47D and ZR-75-1, respectively; Supplementary Fig. 2) the numbers were 4,924 (50.6%) and 4,805 (49.4%; Supplementary Table 1). We found a statistically significant excess of interaction peaks that were common to all four breast cancer cell lines ($N = 62$, permutation test $P = 0.0003$, Fig. 1c) and all five breast cell lines ($N = 53$, permutation test $P < 0.0001$ Fig. 1d). We also found an excess of interaction peaks that were exclusive to the lymphoblastoid cell line ($N = 304$, permutation test $P < 0.0001$) suggesting that at least a subset of interaction peaks show cell-type-specificity. Comparing the cell lines according to receptor type, the interaction peaks were marginally more similar within the two ER+ cell lines (Jaccard similarity coefficient = 0.18) and the two ER− cell lines (Jaccard similarity coefficient = 0.13) than between them (Fig. 1e).

Representative examples of loci that demonstrated cell-type-specific activity are shown in Fig. 2. At several of the ER+ risk loci, including the 10q26.13-rs2981579 (FGFR2) and 14q13.3-

rs2236007 (PAX9) risk loci, we observed interaction peaks that were restricted to the two ER+ breast cancer cell lines and the normal breast epithelial cell line (Fig. 2a and c). In both these examples, the transcription start site (TSS) of the target gene maps within the capture region and forms interaction peaks with specific HindIII fragments that map several hundred kb from the capture region. These distal fragments co-localise with DNase I hypersensitive sites, CTCF, FOXA1, GATA3 and/or ERα binding sites in T-47D cells and in both of these examples the orientation of the CTCF binding sites is towards the captured locus (Fig. 2b, d, e)[22].

There were, however, many exceptions to the pattern of ER+ risk loci forming interaction peaks in the ER+ breast cancer cell lines and Bre80s. The 11p15.5-rs3817198 risk locus, which is associated with ER+ breast cancer forms multiple interaction peaks in the ER− breast cancer cell lines, but not in the ER+ breast cancer cell lines or in Bre80 (Fig. 3a, Supplementary Data 2) and the 6q25.1-rs2046210 (ESR1) locus, which has been shown to be preferentially associated with ER− breast cancer[20,23] forms interaction peaks in the ER+, but not the ER− breast cancer cell lines (Fig. 3b, Supplementary Data 2).

**Defining putative target genes.** We defined putative target genes as genes that mapped within, or *in cis* (≤5 Mb) to, a captured region and for which the TSS mapped to an interacting fragment in at least two cell lines (Methods). On this basis, we were able to assign 110 putative target genes to 33 (64.7%) of the 51 loci (Table 2, Supplementary Data 4); 94 were protein-coding and 16 were non-coding RNAs. The number of genes per locus, for these 33 loci ranged from one (13 loci) to 19 (11q13.1-rs3903072 locus) with a median of two. The distance between the published risk SNP and the TSS of the CHi-C target gene ranged from 1 kb (KCNN4) to more than 4 Mb (3p26.1-rs6762644 with CAV3, RAD18 and SETD5; 11q13.1-rs3903072 with FADD) with a median of 135 kb (individual distances between risk SNPs and CHi-C target genes are given in Supplementary Data 5). Amongst

our 51 informative risk loci there were 24 (at 12 chromosomal regions) that mapped within five Mb of another locus (Supplementary Data 1). We observed interaction peaks between adjacent loci at eight of these chromosomal regions and were able to

potentially assign target genes to three additional loci on the basis of interaction peaks with the adjacent locus (Table 2). These loci were 8q21.11-rs6472903 (*HNF4G* and *PEX2*), 9q31.2-rs10759243 (*KLF4*) and 14q24.1-rs999737 (*ZFP36L1*); a representative

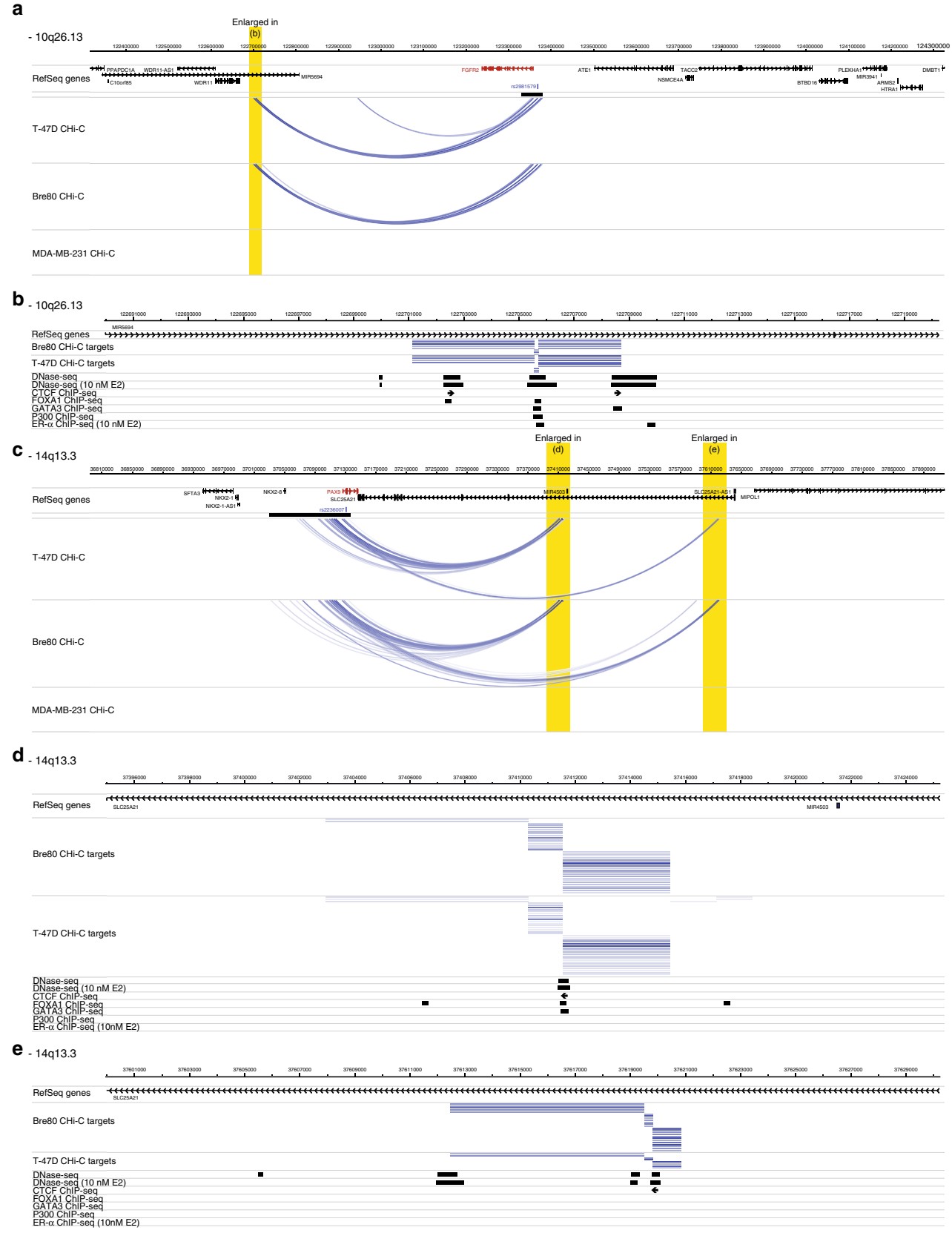

example, showing interaction peaks within and between adjacent loci at 11q13, is shown in Fig. 4. For the interaction peaks with the longest range gene targets (3p26.1-rs6762644: *CAV3*, *LINC00312*, *LMCD1*, *c3orf32*, *RAD18*, *SETD5*, 4q24-rs9790517: *CENPE*, 8q24.21-rs13281615: *CCDC26*, 11q13.1-rs3903072: *CCND1*, *FADD*), we aligned our data with topologically associated domains (TADs) generated in Human Mammary Epithelial Cells (HMEC)[24]. At each of these loci we observed interaction peaks between captured fragments and target gene(s) mapping within the same TAD, but also, less frequently, with target gene(s) mapping to a different TAD (Supplementary Fig. 3).

To determine whether the target genes selected using a CHi-C approach differ from those selected simply on the basis of proximity to the GWAS risk SNP ("nearest gene approach"), we compared the two approaches directly. At 15 of the 51 loci included in our analysis, we were unable to assign target genes (i.e. there were no TSS directly or indirectly involved in statistically significant interaction peaks). Of the 36 loci at which we were able to assign at least one target gene directly ($N = 33$) or indirectly ($N = 3$) there were 24 at which the nearest gene was either the only CHi-C target gene ($N = 9$; Table 2) or one of several CHi-C target genes ($N = 15$; Table 2). There were, however, 12 loci at which our data implicated genes other than the nearest gene; these loci included 13q13.1-rs11571833 (CHi-C gene: *PDSB5*, nearest gene: *BRCA2*), 14q24.1-rs2588809 and rs999737 (CHi-C gene: *ZFP36L1*, nearest gene: *RAD51B*) and 16q12.2-rs17817449 and rs11075995 (CHi-C genes: *CRNDE*, *IRX5*, *IRX3*, *LOC100996*, nearest gene: *FTO*).

**CHi-C target genes and eQTL analyses.** To assess the likelihood of our putative target genes having a causal role in breast cancer aetiology, we first carried out eQTL analyses using the published risk SNPs (or a close proxy, $r^2 > 0.8$) and RNA-Seq data from the Cancer Genome Atlas[25] (TCGA) adjusted for matched DNA methylation and somatic copy-number alterations. Many of risk loci we included have been shown to be associated with breast cancer risk overall, albeit with evidence that the association may differ in magnitude between ER+ and ER− cancers for some[26]. Accordingly, we carried out eQTL analyses for all cancers combined ($N = 547$) and then stratified by ER status (ER+ $N = 415$, ER− $N = 95$, ER unknown $N = 37$). There were 9 loci (26 protein-coding genes) at which there were no suitable proxies, and levels of expression of 18 of our putative target genes (*KRTPA5-5*, *KRTAP5-6* and 16 non-coding RNAs) were too low for analysis. eQTL analysis of the remaining 26 loci (69 protein-coding genes) identified 22 SNP-gene combinations that were nominally significant ($t$-test $P < 0.05$) in all, ER+ or ER− breast cancers (Supplementary Data 6), nine of which remained significant after taking account of multiple testing (FDR adjusted $t$-test $P < 0.1$, Table 3). Comparing these eQTLs with "nearest genes", three were nearest genes and six were not. Including all nearest genes (regardless of whether they were also a CHi-C

target gene) in our eQTL analysis we found two additional SNP-gene combinations that were not captured by our CHi-C analysis; rs4808801 was associated with levels of expression of *ELL* in all cancers and ER+ cancers and rs8170 was associated with levels of expression of *ANKLE* in all cancers (FDR adjusted $t$-test $P < 0.1$).

Several of the CHi-C target gene eQTLs were consistent with previous reports including *IGFBP5* at 2q35-rs13387042[6,27], *COX11* at 17q22-rs6504950[28] and *LRRC25* at 19p13.11-rs4808801[29]. Novel eQTLs included genes that mapped within the capture region, proximal to the reported risk SNP such as *CDCA7* at 2q31.1-rs1550623, *SSBP4* at 19p13.11-rs4808801 and *MRPL34* at 19p13.11-rs8170, as well as genes that mapped several hundred kb from the reported SNP including *IRX3* at 16q12.2-rs17817449 and *ZFP36L1* at 14q24.1-rs2588809. At 11q13.1-rs3903072, eQTL analyses support multiple putative target genes of which *SNX32*, *CTSW* and *CFL1* map within the capture region but, intriguingly, *FADD* and *CCND1* map at a distance of approximately 4 Mb from rs39030702 (Fig. 4). Both *FADD* and *CCND1* map to a region of chromosome 11 that is frequently subject to amplifications and copy-number gains in breast cancer (*FADD* and *CCND1* map to regions of copy-number gain in 20–30% of Metabric[30] and TCGA samples), raising the concern that this long-range eQTL association might be influenced by these samples. Excluding samples with genomic copy-number gains from the analysis, however, strengthened the association between 11q13.1-rs3903072 and *FADD* ($t$-tests: all samples $P_{ER+} = 0.01$, excluding 119 samples with copy-number gains $P_{ER+} = 0.004$; Fig. 5a, c), but not *CCND1* ($t$-tests: all samples $P_{ER+} = 0.04$, excluding 130 samples $P_{ER+} = 0.05$: Supplementary Fig. 4).

**CHi-C target genes and disease-specific survival (DSS).** To our knowledge, only one of the risk SNPs we included has been associated with disease prognosis (16q12.1-rs3803662 and *TOX3*[31]); this may reflect a fundamental difference between the genetics of predisposition and prognosis or a relative lack of power for observational studies of outcome in which detailed information on treatment is generally lacking. As any individual regulatory variant may only explain a small proportion of the total variance in gene expression, however, we looked directly for an association between levels of expression of our putative target genes and patient outcome in the Metabric breast cancer cohort[30]. Given the profound effect of ER status on outcome, we performed survival analyses on ER+ and ER− subpopulations separately. Of the 97 putative target genes (94 protein-coding, 3 non-coding RNAs) for which levels of expression were available, 32 (33%) were associated with DSS in individuals with ER+ disease; none was associated with DSS in ER− disease (FDR adjusted trend test $P < 0.1$; Supplementary Data 7). Comparing these 32 genes with those for which we found eQTL associations in ER+ cancers (nominal $P < 0.05$) there were six that were common to both groups (*CFL1*, *FADD*, *MRPL34*, *IGFBP5*, *IRX3*, *ZFP36L1*). In addition, there was a highly significant association

**Fig. 2** Interaction peaks at 10q26.13 and 14q13.3 in T-47D, Bre80 and MDA-MB-231 cell lines. Interaction peaks are shown in a looping format; interaction peaks in ZR-75–1, BT-20 and GM06990 are not shown but are available online (Methods). Interaction peaks between two captured fragments are red, interaction peaks between one captured fragment and one non-captured fragment are blue. Intensity of individual interactions are proportional to -log₂(P_FDR). Capture regions are shown as black bars; data are aligned with genomic coordinates (hg19) and RefSeq genes. Target genes (i.e. the subset at which an interaction peak co-localises with the TSS) are shown in red. The location of the published risk SNP is also shown. **a** 10q26.13-rs2981579 (*FGFR2*) locus. Interaction peaks originating from the capture region and co-localising with the *FGFR2* TSS, interact with a region ~650 kb centromeric to the locus (highlighted in yellow) in T-47D and Bre80, but not MDA-MB-231. **b** Interaction peaks (shown in blue) at this region co-localise with DNase I hypersensitive sites, CTCF, p300, FOXA1, GATA3 and ERα ChIP-Seq peaks in T-47D cells. The orientation of CTCF peaks is indicated by the direction of the arrow. **c** 14q13.3-rs2236007 (*PAX9*) locus. Interaction peaks originating from the capture region and co-localising with the *PAX9* TSS, interact with two regions ~300 and 500 kb telomeric to the locus (highlighted in yellow) in T-47D and Bre80, but not MDA-MB-231. Scale bar, 80 kb (**d**) and **e** Interaction peaks at these regions co-localise with DNase I hypersensitive sites, CTCF, FOXA1 and GATA3 ChIP-Seq peaks in T-47D cells. The orientation of CTCF peaks is indicated by the direction of the arrow

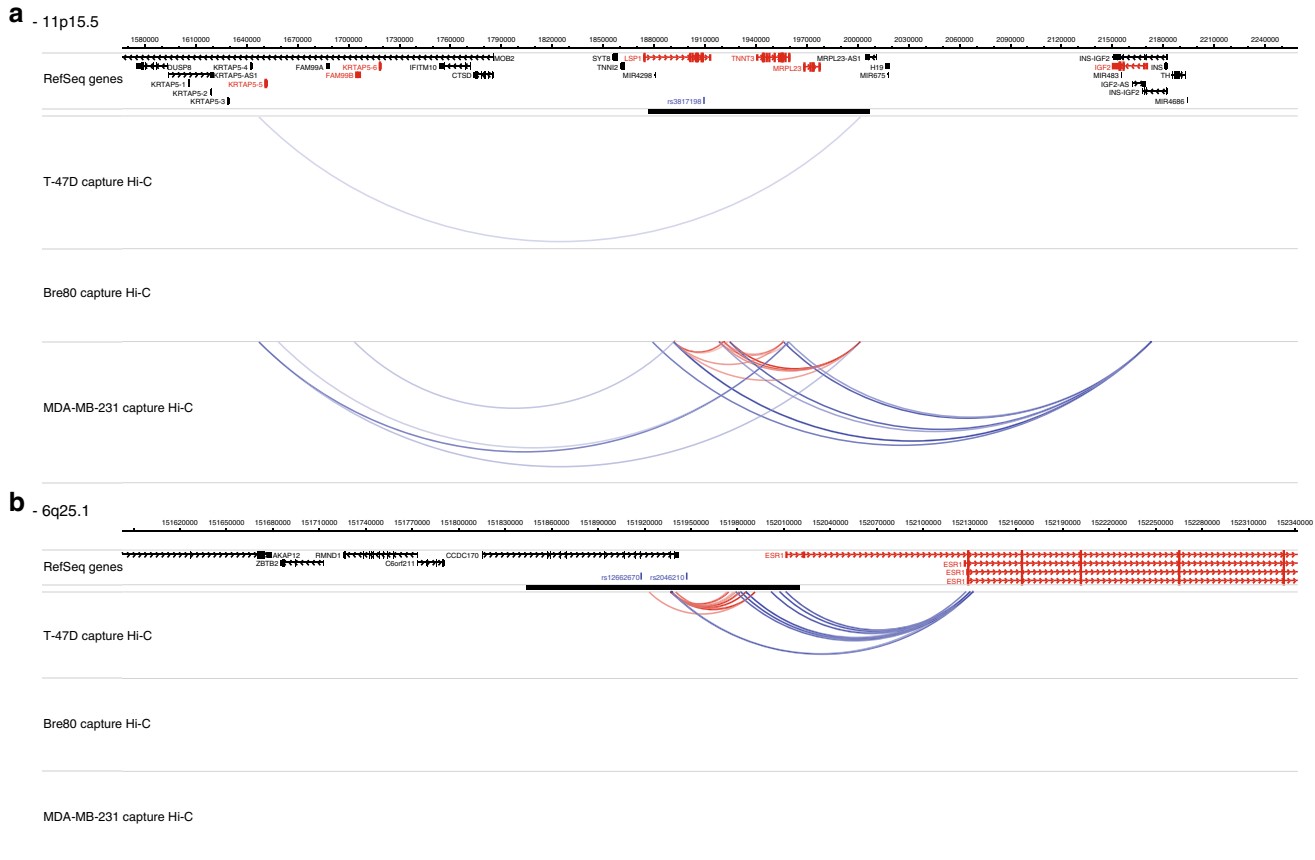

**Fig. 3** Interaction peaks at 11p15.5 and 6q25.1 in T-47D, Bre80 and MDA-MB-231 cell lines. Interaction peaks and genomic features are as described in Fig. 2. **a** 11p15.5-rs3817198 (*LSP1*) locus. At this locus, which is associated with ER+ disease there are multiple interaction peaks targeting *KRTAP5-5* (~300 kb centromeric), *LSP1* (within the capture region) and *IGF2* (~ 250 kb telomeric) in the ER− cell line MDA-MB-231 but just a single IP targeting *KRTAP5-5* in the ER+ cell line T-47D and none in Bre80. **b** 6q25.1-rs2046210 (*ESR1*) locus. At this locus, which is associated with predominantly ER− disease, there are multiple interaction peaks originating from the capture region, overlapping the *ESR1* promoter in the ER+ breast cancer cell line T-47D, but not in the ER− cells breast cancer cell line MDA-MB-231

between levels of expression of *CDCA7* and DSS (trend test $P = 1.22 \times 10^{-8}$), which maps just 7 kb from the reported 2q31.1 risk SNP (rs1550623), but while there was a robust eQTL association between rs1550623 and *CDCA7* in all cancers (trend tests: nominal $P = 0.007$, FDR adjusted $P = 0.09$) there was no association in ER+ cancers alone (both $P > 0.1$). We also observed highly significant associations (FDR adjusted trend test $P < 0.005$) for five genes that were excluded from eQTL analysis due to a lack of suitable tag SNP (*CENPE* at 4q24, *TPCN2* and *ORAOV1* at 11q13.3, *PDS5B* at 13q13.1 and *SLC4A7* at 3p24.1: Supplementary Fig. 5)

**CHi-C target genes and somatic mutations in cancer genes**. Finally, we compared our CHi-C putative target genes with the list of 727 cancer genes compiled by Nik-Zainal and colleagues in their analysis of whole-genome sequences of 560 breast cancers[32]. The 94 protein-coding CHi-C target genes are highly enriched for these cancer genes (14 observed, Hypergeometric $P = 2.02 \times 10^{-6}$) and include well-documented cancer genes (*CCND1*, *CDKN2A*, *CDKN2B*, *MYC*, *MAP3K1*, *ESR1* and *FGFR2*), as well as relatively uncharacterised examples (*TET2*, *KLF4*, *MLLT10*, *FADD*, *TBX3*, *PAX9* and *ZFP36L1*).

Combining the somatic mutation data with the eQTL and DSS analyses, there were 48 CHi-C target genes mapping to 32 loci for which there was orthogonal support from at least one additional source and six genes mapping to six loci for which there was support from at least two additional sources (Table 4). For four of

these, *CDCA7*, *FADD*, *ZFP36L1* and *MRPL34*, levels of expression were associated with both SNP genotype and DSS (Table 4) and we were able to assess whether high (or low) levels of expression were similarly associated with risk and poor outcome. For *FADD* the associations are inconsistent; the rare allele of 11q13.1-rs3903072 is associated with higher levels of expression (Fig. 5a) and lower risk[26], but higher levels of expression are associated with poor outcome (Fig. 5b). In addition, the strong influence of copy-number gains on levels of expression of *FADD* confounds both eQTL and DSS analyses with opposite effects; excluding 119 ER+ cancers with copy-number gains strengthens the eQTL association in TCGA (Fig. 5c), excluding 345 such samples from the analysis of outcome in Metabric abrogates the association with DSS (Fig. 5d), suggesting that samples with copy-number gains at this region may have a poor outcome that is not directly related to levels of expression of *FADD*. Similarly, for *CDCA7* the associations are inconsistent. The risk allele of rs1550623 is the common allele[26]; the common allele is associated with lower levels of *CDCA7* expression (Fig. 5e), but lower levels of expression of *CDCA7* are associated with a better prognosis (Fig. 5f). However, for both 14q24.1-rs2588809 (*ZFP36L1*) and 19p13.1-rs8170 (*MRPL34*) the rare alleles are associated with lower levels of expression (Fig. 5g, i) and higher risk[26]; lower levels of expression are also associated with a poor outcome (Fig. 5h, j) consistent with these genes acting as tumour suppressors influencing both predisposition and outcome similarly.

**Table 2 Risk loci which formed interaction peaks directly (N = 33) or via an adjacent risk locus (N = 3) with 110 target genes**

| Locus | SNP(s) | CHi-C target genes | Nearest gene | Agrees |
|---|---|---|---|---|
| 1p36.22 | rs616488 | APITD1; DFFA, **PEX14**; PGD | PEX14 | √+ |
| 1p13.2 | rs11552449 | OLFML3; HIPK1 | DCLREB1 | X |
| 2q31.1 | rs2016394 | CDCA7; **DLX2**; DYNC1I2 | DLX2 | √+ |
| 2q31.1 | rs1550623 | **CDCA7**; DLX2 | CDCA7 | √+ |
| 2q35 | rs13387042 | IGFBP5 | *LINC01921*, TNP1 | X |
| 2q35 | rs16857609 | IGFBP5; RPL37A | *DIRC3*, TNS1 | X |
| 3p26.1 | rs6762644 | BHLHE40; CAV3; *LINC00312*; LMCD1; c3orf32; RAD18; SETD5 | ITPR1 | X |
| 3p24.1 | rs4973768 | NGLY1; OXSM; **SLC4A7** | SLC4A7 | √+ |
| 4q24 | rs9790517 | CENPE; PPA2; **TET2** | TET2 | √+ |
| 5q11.2 | rs889312 | **MAP3K1** | MAP3K1 | √ |
| 6p23 | rs204247 | **RANBP9** | RANBP9 | √ |
| 6q25.1 | rs12662670, rs2046210 | ESR1 | CCDC170 | X |
| 8q21.11 | rs6472903 | **HNF4G/PEX2** (adj) | CASC9, HNF4G | √+ |
| 8q21.11 | rs2943559 | **HNF4G**; PEX2; | HNF4G | √+ |
| 8q24.21 | rs13281615 | *CCDC26  CASC11*  **MYC** | *CASC8, CASC21, POU5F1B*, MYC | √+ |
| 8q24.21 | rs11780156 | *CCDC26*; *CASC11*, **MYC** | *PVT1*, MYC | √+ |
| 9p21.3 | rs1011970 | CDKN2A; **CDKN2B**; MTAP | CDKN2B | √+ |
| 9q31.2 | rs10759243 | **KLF4** (adj) | KLF4 | √ |
| 9q31.2 | rs865686 | **KLF4** | KLF4 | √ |
| 10p12.31 | rs7072776, rs11814448 | BMI1; COMMD3; *LOC100499489*; *MIR1915*, c10orf114; **MLLT10**; c10orf140 | DNAJC1, MLLT10 | √+ |
| 10q22.3 | rs704010 | **ZMIZ1** | ZMIZ1 | √ |
| 10q26.13 | rs2981579 | **FGFR2** | FGFR2 | √ |
| 11p15.5 | rs3817198 | *FAM99B*, KRTAP5-6; IGF2; KRTAP5-5; MRPL23, *SNORD131*; TNNT3; **LSP1**, *LINC01150*. | LSP1 | √+ |
| 11q13.1 | rs3903072 | DKFZp761E198, *MIR1234*; OVOL1; C11orf68, DRAP1; CCDC85B; CFL1; CTSW; FIBP; FOSL1; KAT5; MUS81, EFEMP2; RNASEH2C; SART1; **SNX32**; TSGA10IP; CCND1, FADD | SNX32 | √+ |
| 11q13.3 | rs554219, rs78540526, rs75915166 | **CCND1**, *LINC01488*; ORAOV1; FADD; *LOC338694*; *MIR3164*; MRGPRF; MRPL21; IGHMBP2; MYEOV; TPCN2 | *LINC01488*, CCND1 | √+ |
| 12q24.21 | rs1292011 | **TBX3** | TBX3 | √ |
| 13q13.1 | rs11571833 | PDS5B | BRCA2 | X |
| 14q13.3 | rs2236007 | **PAX9** | PAX9 | √ |
| 14q24.1 | rs2588809 | ZFP36L1 | RAD51B | X |
| 14q24.1 | rs999737 | ZFP36L1(adj) | RAD51B | X |
| 16q12.2 | rs17817449, rs11075995 | *CRNDE*, IRX5; IRX3; *LINC02169* | FTO | X |
| 17q22 | rs6504950 | **STXBP4**, COX11; TOM1L1 | STXBP4 | √+ |
| 19p13.1 | rs8170, rs2363956 | ANO8, GTPBP3; DDA1; MRPL34; NR2F6 ; USE1; OCEL1 | BABAM1, ANKLE1 | X |
| 19p13.11 | rs4808801 | LRRC25; SSBP4, ISYNA1; LSM4; MRPL34; PGPEP1, GDF15, *MIR3189*; UPF1 | ELL | X |
| 19q13.31 | rs3760982 | **KCNN4** | KCNN4 | √ |
| 22q13.1 | rs6001930 | *LOC101927257* | MLK1 | X |

Where TSS for two or more target genes map to a single HindIII fragment, the genes are separated by a comma. Non-coding RNAs (long non-coding RNAs, microRNAs and small nucleolar RNAs) are indicated in green. There were three loci at which the target gene is assigned indirectly on the basis of interaction peaks with an adjacent locus; these are indicated by (adj). Defining nearest gene; for SNPs that map within a gene (UTR, exons or introns) this gene is considered to be the nearest gene, for SNPs that do not map within a gene, nearest gene is assigned based on the location of the nearest TSS according to RefSeq genes (GRCh37/hg19). Where the nearest gene is a non-coding RNA, the nearest protein-coding gene is also given. CHi-C target genes that are also the nearest gene are indicated in bold. CHi-C targets and the nearest gene are compared in the "Agrees" column; √ CHi-C data were consistent with the nearest gene being the sole target gene, √+ CHi-C data were consistent with the nearest gene being one of several target genes, X CHi-C data support a gene other than the nearest gene as a target

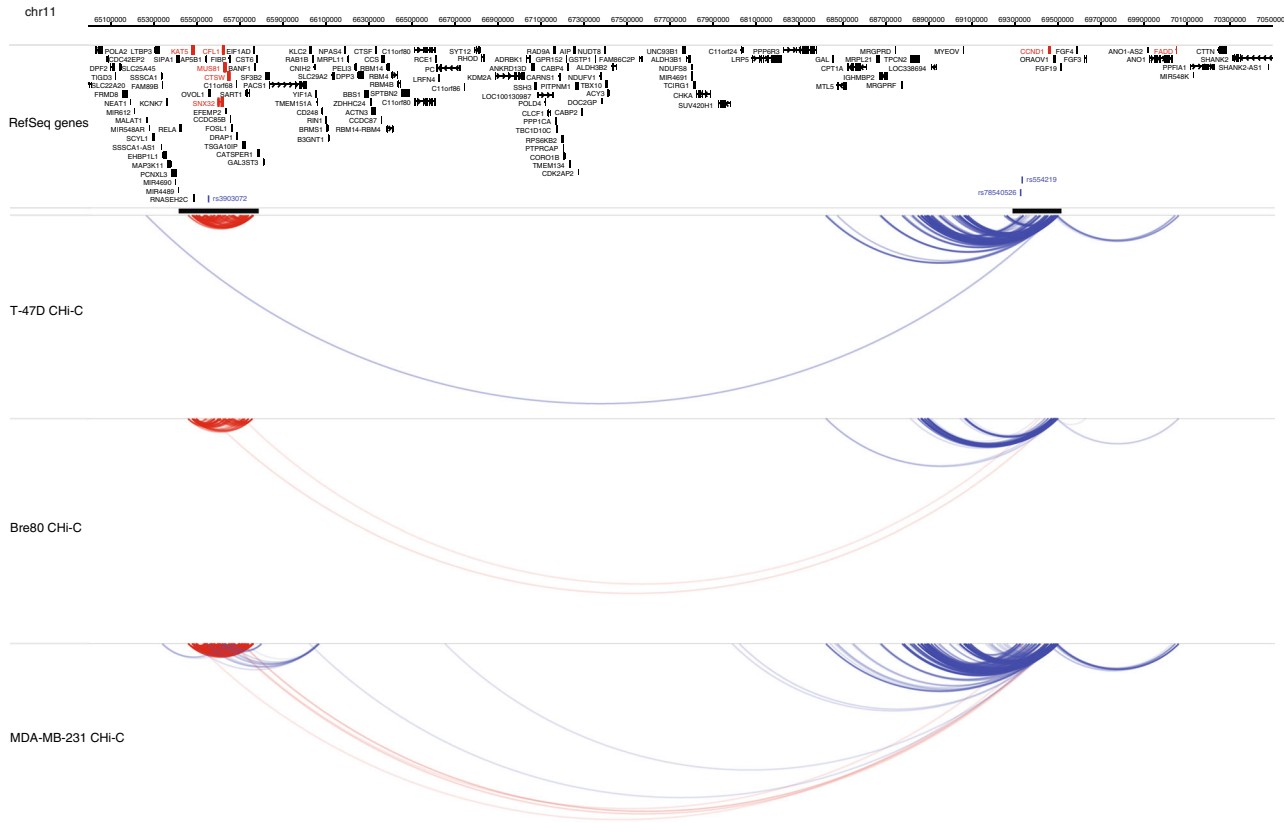

**Fig. 4** Interaction peaks at 11q13.1 and 11q13.3 in T-47D, Bre80 and MDA-MB-231 cell lines. Interaction peaks and genomic features are as described in Figs 2 and 3, with the exception that only target genes that are eQTLs are in red. In addition to the local within capture and *in cis* interaction peaks at each of these loci there are long-range (>4 Mb) interaction peaks between the two risk loci in both Bre80 and MDA-MB-231 cell lines. Target genes that are eQTLs for the 11q13.1 risk SNP (rs3903072) are *CFL1, CTSW, KAT5, MUS81, SNX32, CCND1* and *FADD*. Three interactions are omitted for clarity (see Methods)

## Discussion

The purpose of this analysis was to identify target genes at 63 breast cancer GWAS risk loci using an unbiased, high-resolution chromosome association method, CHi-C and evaluate this method in comparison to a simple "nearest gene" approach. We were able to assign 110 putative target genes to 33 loci; 94 were protein-coding and 16 were non-coding RNAs. We used three publicly available data sources to assess support for our CHi-C target genes as having a causal role in breast cancer aetiology. In eQTL analyses, we identified 22 SNP-gene combinations that were nominally significant (*t*-test $P < 0.05$) in all, ER+ or ER− breast cancers in TCGA. DSS analyses of ER+ breast cancers in the Metabric cohort supported 32 CHi-C target genes (FDR adjusted trend test $P < 0.1$) and 14 were listed in 727 cancer genes compiled by Nik-Zainal and colleagues. In all data sources combined there were support for 48 CHi-C target genes mapping to 32 loci from at least one additional source and there was support for six genes mapping to six loci from at least two additional sources. These data suggest that a substantial proportion of the CHi-C putative target genes are likely to influence breast cancer risk and warrant further investigation.

However, amongst the 63 risk loci that we investigated there were 12 at which we detected no interaction peaks at all. This may, in part, be a consequence of our methodology as 3C-based techniques are not considered reliable for detecting interactions over distances of less than 10 kb[33]; at three of these 12 loci (4q34.1-rs6828523 (*ADAM29*), 10q21.2-rs10995190 (*ZNF365*) and 16q12.1-rs3803662 (*TOX3*) the TSS of a nominated target gene mapped within 10 kb of the reported SNP. Similarly, for the 15 loci at which the interaction peaks, we detected did not include

direct, or indirect, interactions with the promoter of a RefSeq gene, there were four at which the TSS of the proposed target gene mapped within 10 kb (6p25.3-rs11242675 (*FOXQ1*), 22q12.1-rs17879961 (*CHEK2*)) or 20 kb (5p15.33-rs10069690 (*TERT*), 22q12.1-rs132390 (*EMID1*)) of the reported SNP (Supplementary Data 2). In any analysis, there is a trade-off between type I and type II errors. By using a rigorous threshold (FDR adjusted outlier test $P < 0.01$) for calling an interaction peak "significant" we will have minimised false positives, but we may also have missed potentially important low frequency interactions. Finally, we may have missed important target genes by using a restricted set of cell lines that will only capture interaction peaks between regulatory elements and genes that are expressed in breast epithelial cells. At the other extreme, there were several loci mapping to gene-rich regions (particularly 11q13 and 19p13), at which we observed interaction peaks with multiple putative target genes some of which mapped to the same HindIII restriction fragment as another target gene (Table 2). Reducing the size of the average restriction fragment, by using an enzyme that cuts more frequently would provide greater resolution, but it is clear that CHi-C cannot resolve interaction peaks at the TSS of putative target genes that map within a few hundred base pairs of each other.

The other metrics that are frequently used for defining putative target genes are nearest gene, or nearest plausible gene, and eQTL analyses. While in many cases our analyses support the nearest gene or the nearest plausible gene the limitations to this approach are obvious; there are many examples of long-range interactions between regulatory elements and target genes that bypass more proximal putative target genes[34–36]. Comparing CHi-C with a

**Table 3 Nine CHi-C putative target genes that were statistically significant eQTLs (FDR adjusted $P < 0.1$)**

| Cytoband | SNP | Proxy | Gene | | All cancers | | ER+ cancers | | ER− cancers | |
|---|---|---|---|---|---|---|---|---|---|---|
| | | | Nearest | CHi-C target | $P$ | $P_{adj}$ | $P$ | $P_{adj}$ | $P$ | $P_{adj}$ |
| 2q31.1 | rs1550623 | | CDCA7 | **CDCA7** | 0.007 | 0.087 | 0.511 | 0.666 | 0.330 | 0.892 |
| 11q13.1 | rs3903072 | | SNX32 | CTSW | 0.006 | 0.087 | 0.064 | 0.326 | 0.001 | 0.101 |
| 11q13.1 | rs3903072 | | SNX32 | **SNX32** | 0.007 | 0.087 | 0.032 | 0.268 | 0.036 | 0.506 |
| 14q13.3 | rs2236007 | rs1018464 | PAX9 | **PAX9** | 0.003 | 0.066 | 0.054 | 0.317 | 0.248 | 0.854 |
| 14q24.1 | rs2588809 | | RAD51B | ZFP36L1 | 0.079 | 0.380 | 0.004 | 0.091 | 0.256 | 0.854 |
| 17q22 | rs6504950 | rs9902718 | STXBP4 | COX11 | 0.002 | 0.059 | 0.001 | 0.032 | 0.403 | 0.892 |
| 19p13.11 | rs8170 | rs34084277 | BABAM1, ANKLE1[*] | MRPL34 | 0.001 | 0.059 | 0.011 | 0.173 | 0.131 | 0.829 |
| 19p13.11 | rs4808801 | | ELL[*] | LRRC25 | 0.009 | 0.092 | 0.004 | 0.091 | 0.768 | 0.954 |
| 19p13.11 | rs4808801 | | ELL[*] | SSBP4 | 0.002 | 0.059 | 0.0002 | 0.016 | 0.475 | 0.892 |

[*]In an analysis that included all genes that are nearest genes, regardless of whether they were also a CHi-C target gene, rs4808801 was also associated with expression of *ELL* (FDR adjusted $P = 0.05$ for all cancers and $P = 0.04$ for ER+ cancers) and rs8170 was also associated with expression of *ANKLE1* (FDR corrected $P = 0.05$ for all cancers)
$P$ $P$-value (1df $t$-test) per allele association with gene expression, adjusted for methylation and copy number, $P_{adj}$ $P$ value further adjusted for multiple testing

nearest gene metric for assigning putative target genes to risk loci, our data were informative at 36 (57%) of the 63 loci we selected for analysis and at 27 (43%), our data implicated genes in addition to, or other than, the nearest gene. Notably, our data implicate several protein-coding genes and non-coding RNAs that map at distances of more than 1 Mb from the published risk SNP (i.e. outside the limit of many eQTL analyses). While the presence of these long-range interactions may inform future follow up studies, they do not exclude effects that are more local to the risk loci. In their functional annotation of the human genome, the ENCODE consortium estimated that the average number of TSSs that interact with any given distal element is 2.5[37] and regulatory variants that map to such elements may influence absolute or relative levels of expression of multiple genes. Of the six genes for which there was support from at least two additional data sources, neither *ZFP36L1* (which maps 600 kb from rs2588809) nor *FADD* (mapping 4.5 Mb from rs3903072) would have been selected by a nearest gene metric supporting the use of CHi-C as a means of identifying putative target genes that map several hundred kb or even Mb from the risk locus.

eQTL analyses provide an intuitive approach to the process of identifying putative target genes. However, implicit in eQTL analyses of breast cancer or normal breast tissue is an assumption of a model in which a breast cancer GWAS locus influences risk by altering steady-state expression of a gene that is transcribed in normal or malignant breast tissue; this may not be true for a substantial minority of loci. For this reason, a CHi-C-based approach which detects "permissive" interaction peaks (as well as "instructive" interaction peaks)[33,38] may have benefits over an eQTL-based approach by allowing the identification of putative target genes that are poised for expression at a particular stage of differentiation or in response to external stimuli such as hormones or DNA damage.

The variants detected by GWAS are common variants with small effects (ORs are typically <1.2) and any individual risk SNP will usually only explain a small proportion of variance in levels of expression of a target gene. For example, the association between 11q13.1-rs3903072 and *FADD* is weak in all ER+ cancers ($t$-test: nominal $P = 0.01$); excluding ER+ cancers with copy-number gains reduces the variance in levels of expression of *FADD* and increases the proportion of variance explained by rs3903072 ($t$-test: nominal $P = 0.004$). Given the small effects of individual variants, eQTL approaches based on current data sets of a few hundred samples lack power. To limit the penalty for multiple testing, most eQTL analyses are restricted to genes within a 1 Mb window of the risk SNP or a proposed causal

variant. In our eQTL analysis we used our CHi-C results to restrict our gene set to 69 protein-coding genes. Despite this our eQTL analysis probably lacked power, particularly for the stratified analyses where there were just four ER+ eQTLs and no ER− eQTLs that were significant after taking account of multiple testing. Indeed *IGFBP5*, *KLF4*, *CFL1*, *CCND1* and *IRX3* are all fairly compelling putative target genes with nominal associations for which the adjusted eQTL $P$-values were non-significant ($t$-tests: all nominal $P_{ER+} < 0.05$, all FDR adjusted $P_{ER+} > 0.1$).

For several of the risk loci that we included, functional annotation studies have been previously reported on a locus-by-locus basis[27,28,39–51] and target genes have been inferred by a combination of proximity, eQTL analysis and testing for looping interactions on a candidate basis using 3 C. Our CHi-C targets are consistent with many of these[27,28,43,46,47], but may implicate *CENPE* in addition to *TET2* at 4q24[40] and *MRPL34* in addition to *ABHD8* or *ANKLE1* at 19p13.1[51]. The other notable feature of our data is the frequency with which we observed interaction peaks between adjacent loci several of which map megabase distances apart. This feature, and our observation of an eQTL between rs3903072 and both *CFL1* and *FADD*, which map 4.5 Mb apart, suggests that the number of target genes may be less than the number of reported risk loci albeit with, potentially, multiple co-regulated target genes at some loci.

Overall it is difficult to evaluate our list of putative target genes when fully understanding the mechanisms by which a given gene influences cancer risk are often complex and require many years' work. It seems likely, however, that the first stage of this process will be short-listing candidates for follow up studies. On that basis, we would argue that a high-throughput CHi-C analysis can contribute to on-going efforts to functionally annotate GWAS risk loci and that CHi-C target genes that are supported by additional data sources are strong candidates for in depth functional follow up studies.

## Methods

**Target enrichment array design**. 74 SNPs mapping to 68 GWAS risk loci were selected based on all available published GWAS and replication studies as of 31/01/2015. Capture regions were defined as the region that included all SNPs that were correlated ($r^2 \geq 0.2$) with the published SNP based on 1000 Genomes pilot data (http://www.1000genomes.org/; Supplementary Data 1). Biotinylated 120-mer RNA baits were designed to target both ends of the HindIII restriction fragments that mapped within these capture regions using Agilent eArray software (Agilent, Santa Clara, CA, USA), using 2 × tiling, moderately stringent repeat masking and maximum performance boosting options.

**Cell culture and formaldehyde crosslinking**. T-47D, ZR-75-1, BT-20 and MDA-MB-231 cell lines were obtained from ATCC (Middlesex, UK), GM06990 cells were

supplied by Coriell Cell Repositories (Coriell Institute for Medical Research, New Jersey, USA). Normal Bre80 TERT-immortalised mammary epithelial cells Bre80-Q-TERT (Bre80) were kindly provided by Prof Georgia Chenevix-Trench (Queensland Institute of Medical Research, Brisbane, Queensland, Australia). Cell lines were authenticated using STR genotyping and were regularly tested for

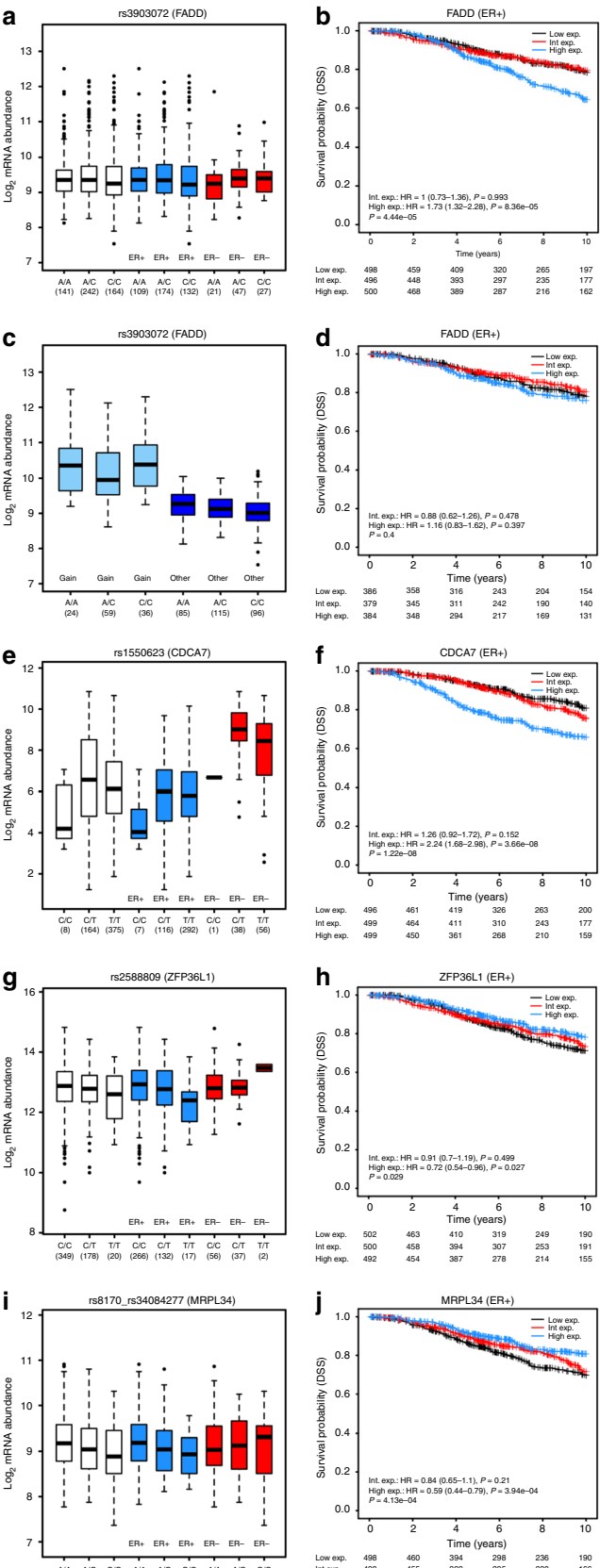

mycoplasma contamination. Bre80 cells were grown in DMEM/F12 with phenol red (Gibco, Life Technologies) supplemented with 5% horse serum, 10 μg/ml insulin, 0.5 μg/ml hydrocortizone, 20 ng/ml epidermal growth factor, 100 ng/ml cholera toxin, 50 U/ml penicillin and 50 μg/ml streptomycin (Sigma-Aldrich, St. Louis, MO, USA). T-47D and ZR-75-1 were grown in RPMI 1640 (Gibco, Life Technologies) supplemented with 10% foetal bovine serum (FBS, Life Technologies), 50 U/ml penicillin, 50 μg/ml streptomycin (Sigma-Aldrich, St. Louis, MO, USA) and, for T-47D, 10 μg/ml insulin (Sigma-Aldrich, St. Louis, MO, USA). BT-20 cells were grown in EMEM (ATCC, Middlesex, UK) supplemented with 10% FBS, 50 U/ml penicillin, 50 μg/ml streptomycin. MDA-MB-231 cells were grown in DMEM supplemented with 10% FBS, 50U/ml penicillin, 50 μg/ml streptomycin and GM06990 cells were grown in RPMI 1640 supplemented with 15% FBS, 50 U/ml penicillin, 50 μg/ml streptomycin and 2 mM L-Glutamine. Formaldehyde crosslinking of 20 million cells was performed as described by Belton and collea-gues[52] by substituting standard culture media with FBS-free media containing 2% formaldehyde for 5 min at room temperature. Crosslinking was quenched by addition of glycine to a final concentration of 150 mM. Adherent T-47D, ZR-75-1, BT-20, MDA-MB-231 and Bre80 cells were scraped off the culture flask after crosslinking, non-adherent (GM06990) cells were transferred directly to a falcon tube. Cells were washed with cold PBS, snap-frozen in liquid nitrogen and stored at −80 °C before preparation of the Hi-C library.

**Hi-C library generation.** Each cross-linked cell aliquot (~20 million cells) was resuspended in 50 ml of permeabilisation buffer (10 mM Tris-HCl pH8, 10 mM NaCl, 0.2% IGEPAL CA-630 (Sigma-Aldrich, St. Louis, MO, USA), supplemented with complete mini EDTA-free tablets (Roche, Basel, Switzerland) and incubated on ice for 30 min with occasional mixing. T-47D, ZR-75-1 and GM06990 cells were lysed using 10 strokes of a dounce homogeniser. BT-20, MDA-MB-231 and Bre80 cells were lysed by incubating with trypsin (0.25%, Sigma-Aldrich, St. Louis, MO, USA) at 37 °C for 5 min. Trypsin was inactivated by addition of 500 μl FBS. Per-meabilised cells were centrifuged for 6 min at 600x g and washed three times in 1 ml 1.3 × NEBuffer 2 (New England Biolabs, Ipswich, MA, USA). Nuclei were resuspended and chromatin digestion and Hi-C library preparation were carried out as described by van Berkum and colleagues[7] with the following modifications: (i) cells were split into three microcentrifuge tubes instead of five (ii) restriction fragment overhangs were filled in with biotinylated dATP instead of biotinylated dCTP (iii) dGTP was not added to the reaction mixture for the removal of bio-tinylated dATP from unligated ends (iv) an agarose gel size selection step was not included, and (v) after PCR amplification (5–8 cycles) of the Hi-C library-bound streptavidin beads the PCR product was pooled and subjected to target enrichment (below) before paired-end sequencing.

**Target enrichment.** Target enrichment was performed based on the SureSelect protocol (Agilent, Santa Clara, CA, USA), but incorporating the following mod-ifications: (i) Biotinylated Hi-C di-tags bound to streptavidin beads were amplified pre-hybridisation directly from beads using 24 parallel 25 μl PCR reactions with five to eight cycles using Q5 High-Fidelity DNA Polymerase (New England Biolabs, Ipswich, MA, USA) and pre-hybridisation PCR primers: ACACTCTTTCCCTA-CACGACGCTCTTCCGATC*T and CTCGGCATTCCTGCT-GAACCGGCTCTTCCGATC*T. PCR products were pooled and purified using Agencourt Ampure XP beads (Beckman Coulter, Brea, CA, USA) to yield ~750–1300 ng total DNA. 750 ng of library DNA was dried using a speedvac concentrator then resuspended in 3.4 μl of water. (ii) Enriched fragments were amplified post-hybridisation again directly from the streptavidin beads, using 18 parallel 25 μl reactions of five to eight cycles of PCR. PCR products were again pooled and purified using Agencourt Ampure XP beads (Beckman Coulter, Brea, CA, USA). Post-hybridisation PCR primers to the paired-end adaptors were as described in Belton and colleagues[52]

**Fig. 5** Box plots of gene expression according to genotype and Kaplan–Meier plots of disease-specific survival according to levels of expression for *FADD* (11q13), *CDCA7* (2q31.1), *ZFP36L1* (14q24.1) and *MRPL34* (19p13.1). **a** Levels of expression of *FADD* are associated with 11q13.1-rs3903072 genotype in all cancers (P = 0.04) and ER+ cancers (P = 0.01); **b** in ER+ cancers, levels of expression of *FADD* are also associated with disease-specific survival (DSS) (**c**) excluding samples with copy-number gains strengthened the eQTL association in ER+ cancers (P = 0.004) (**d**) but attenuated the association with DSS. **e**, **g**, **i** Levels of expression of *CDCA7*, *ZFP36L1* and *MRPL34* are associated with 2q31.1-rs1550623 genotype in all cancers (P = 0.007), 14q24.1-rs2588809 genotype in ER+ cancers (P = 0.004) and 19p13.1-rs8170 in all cancers (P = 0.001) and ER+ cancers (P = 0.01), respectively. **f**, **h**, **j** In ER+ cancers, levels of expression of all three genes are associated with DSS

### Table 4 Six CHi-C putative target genes for which there was orthogonal support for at least two additional data sources

| Locus | SNP | Gene | eQTL $P_{all}$ | eQTL $P_{ER+}$ | DSS $P_{ER+}$ | 727 cancer genes source |
|---|---|---|---|---|---|---|
| 2q31.1 | rs1550623 | CDCA7 | 0.007 (0.09) | 0.51 (0.67) | $4 \times 10^{-7}$ | |
| 11q13.1 | rs3903072 | FADD | 0.04 (0.28) | 0.01* (0.17) | 0.0009 | Cancer related genes panel |
| 12q24.21 | rs1292011 | TBX3 | 0.28 (0.50) | 0.21 (0.52) | 0.012 | Cancer gene census |
| 14q13.3 | rs2236007 | PAX9 | 0.003 (0.07) | 0.05 (0.32) | 0.20 | Cancer related genes panel |
| 14q24.1 | rs2588809 | ZFP36L1 | 0.08 (0.38) | 0.004 (0.09) | 0.09 | Identified in Nik-Zainal et al 2016 |
| 19p13.11 | rs8170 | MRPL34 | 0.001 (0.06) | 0.01 (0.17) | 0.004 | |

eQTL P values in parenthesis are FDR adjusted, * excluding 119 ER+ cancers with copy-number gains at FADD, P = 0.004

**Next generation sequencing (NGS), mapping and filtering**. A total of 12 target enriched Hi-C libraries (two biological replicates for each of six cell lines) were prepared. Eight of the libraries (all at concentrations >2,500 pM) were sequenced on single flow cell lanes on an Illumina HiSeq2000 (Illumina, San Diego, CA, USA) generating 76 bp paired-end reads. The other four libraries, which were at lower concentrations (330–630 pM), were sequenced on two flow cell lanes each. Casava software (v1.8, Illumina) was used to make base calls; reads failing the Illumina chastity filter were removed before further analysis. Sequences were output in fastq format before mapping against the human reference genome (GRCh37/hg19) generating between 86 and 153 million di-tags with both ends uniquely mapped to the reference genome. Filtering to remove experimental artefacts was carried out using the publicly available Hi-C User Pipeline (HiCUP). Full details of this pipeline are available from Babraham Bioinformatics (http://www.bioinformatics.babraham.ac.uk/). In addition to the standard pipeline, off-target di-tags (defined as di-tags where neither end mapped to one of the capture regions) were removed from the final processed data sets. After excluding invalid pairs[52,53], PCR duplicates and off-target di-tags, the number of valid di-tags ranged from 24 to 71 million. Full details of the number and proportion of excluded di-tags are given in Supplementary Table 2.

**Analysis of Hi-C interaction peaks**. The power of our analysis to detect significant interaction peaks depends on the read density, which in turn depends on the size of the bin or unit of analysis. Given that our purpose was to identify individual target genes, we restricted the analysis to a high-resolution (single HindIII fragment) analysis of valid di-tags generated by ligations between a captured fragment and (i) another captured fragment in cis or (ii) a non-captured fragment in cis, mapping within 5 Mb[6]. We carried out separate analyses for each type of ligation on the basis that the statistical properties of ligations where both ends of the di-tag have been captured (type (i)) will differ from those where just one end has been captured (type (ii)).

To assess the reproducibility of our libraries we calculated Spearman's $\rho$ for each possible combination of HindIII fragments, for each type of analysis ((i) and (ii)) using the two biological replicate libraries for each of the six cell lines. We excluded combinations of HindIII fragments for which there were zero read pairs in both libraries and stratified our analysis on the distance between the two HindIII fragments (0–500 kb, 500 kb–1 Mb, 1 Mb–1.5 Mb, >1.5 Mb). The correlation between duplicates was strongest when both fragments were captured and mapped within 500 kb of each other ($\rho = 0.78$ to $\rho = 0.92$). For fragments separated by distances of >1 Mb (where most of the raw di-tags represent "noise") there was weak or no correlation between replicates (all $\rho < 0.4$); for fragments separated by 500 kb to 1 Mb correlation was moderate ($\rho = 0.53$ to $\rho = 0.77$ when both fragments were captured, and $\rho = 0.33$ to $\rho = 0.59$ when just one fragment was captured; Supplementary Figs. 6 and 7).

There were eight loci annotated by 10 SNPs (5p15.33-rs10069690, 5p15.33-rs7726159 and rs2736108, 11q13.3-rs554219 and rs78540526, 11q13.3-rs75915166, 16q12.2-rs17817449, 16q12.2-rs11075995, 19p13.1-rs8170, 19p13.1-rs2363956), where the capture regions were too close for us to analyse separately. Accordingly, we collapsed these eight regions into four. There was one region at 10q23.1-rs7071985 (82909977-83064943) that failed to generate high numbers of reads in any of the cell lines we assayed. After excluding this region and combining eight regions into four, there were 63 separate loci for analysis (Supplementary Data 1). On our arrays, there were also 1,254 captured HindIII fragments that did not map to known breast cancer risk loci and were not considered further in this study. Three of these fragments comprising the GSTP1 promoter, mapped within 5 Mb of the 11q13.1-rs3903072 capture region and formed interaction peaks with this capture region. For clarity these interaction peaks are excluded from Fig. 4. For the 63 risk loci we generated data sets that comprised all di-tags in both categories (type (i) and (ii)) using the SeqMonk mapped sequence analysis tool (www.bioinformatics.babraham.ac.uk/projects/seqmonk/). Where a captured region mapped within 5 Mb of another captured region we considered HindIII fragments mapping to these two regions as part of a "within capture" (type (i)) analysis.

In common with other "C"-based techniques, our Capture Hi-C methodology includes several steps that will show local differences in efficiency thereby introducing biases in the detection of interaction peaks[35]. To correct for these biases, we used a modification of the procedure described by Sanyal and

colleagues[6,35]. Briefly, our method assumes that some of our captured fragments "fail" and we exclude these; we then used a truncated negative binomial model, which takes account of both the large number of zero counts in the data and allows for overdispersion, to model all ligations for which one end maps an unexcluded captured fragment. R-scripts are available on request. In detail, on the assumption that the majority of trans ligations represent random events, we calculated the total number of trans ligations ($N_T$) made by each of the captured HindIII fragments as a measure of the fragment's "interactability", its propensity to interact with other fragments. The interactability had a bimodal distribution, which we assumed to arise from two components corresponding to low numbers of counts, which we regarded as stochastic noise, and higher numbers of counts, which we regarded as genuine signal. For each cell line and biological replicate a truncated negative binomial distribution, based on the number of di-tags, was fitted to the higher component. By visually inspecting the histogram, it was apparent that the truncation point varied between each cell line and biological replicate. Both the histograms and individual truncation points were used to define an individual threshold, this being the 5% quantile point of the corresponding non-truncated distribution, for each cell line and replicate. All fragments with a total number of trans di-tags below the corresponding threshold value were regarded as noise and were filtered out. This resulted in excluding between 8.6% and 30.0% of the fragments with the lowest number of trans ligations (di-tags). We fitted negative binomial regression models to the filtered data sets, combining data from the two biological replicates for each cell line. We corrected for experimental biases due to differing interactability of fragments by including as a covariate the log_e of the total number of trans ligations ($\ln(N_T)$) for each captured fragment from each biological replicate; for cis ligations within the capture regions we also included a term for interaction products of $\ln(N_T)$ for each of the two ligated fragments in each biological replicate. We corrected for distance between the ligated fragments by including as a covariate the log_e of the distance between the mid-points of the two fragments ($\ln(D)$); to approximate local smoothing we fitted the data in bins each of which contained 1 percentile of the distance range. P-values were obtained by comparing the observed counts to the fitted distributions. For each capture region in each cell line, we controlled the false discovery rate using the method of Benjamini & Hochberg[54]. Supplementary Figs. 8 and 9 show raw read counts aligned to the reported interaction peaks in two libraries ((i) T-47D and (ii) MDA-MB-231 at the 10q26.13-rs2981579 locus (see Fig. 2a) and the 11p15.5-rs3817198 locus (see Fig. 3a). Data were visualised using the WashU Epigenome Browser: http://epigenomegateway.wustl.edu/browser/ and aligned with DNase I and ChIP-Seq data from ENCODE: https://www.encodeproject.org/ (Supplementary Table 3)

**Comparison of interaction peaks between cell lines**. Non-parametric equality tests (Mann–Whitney for two samples, Kruskal–Wallis for multiple samples) were used to test for a difference in the median number of interaction peaks per locus and the median distance between interacting fragments, across cell lines. We tested the probability of an excess of shared interaction peaks among breast cancer cell lines and all breast-specific cell lines using a random sampling (10,000 permutations) and we estimated the similarity according to receptor status (ER+ /ER−) for the breast cancer cell lines using the Jaccard similarity coefficient.

**Allocating putative target genes and nearest genes**. To define a set of putative target genes, we identified all catalogued RefSeq genes (GRCh37/hg19), mapping within, or in cis ($\leq$5 Mb) to a captured region. From these we selected the subset for which the TSS mapped to one end of an interaction peak (an interacting fragment). Given that cancer cell lines are aneuploid, with multiple rearrangements and regions of loss or gain, we further required that the TSS mapped to an interacting fragment in at least two cell lines. For SNPs that mapped to a RefSeq gene (UTR, exon or intron), this gene was considered to be the nearest gene (Table 2). For intergenic SNPs, the nearest gene was determined on the basis of the nearest RefSeq catalogued TSS. Where the nearest catalogued TSS was for a non-coding RNA, this non-coding RNA is listed along with the nearest protein-coding gene (Table 2).

**Aligning CHi-C data with TADs**. In order to align CHi-C data with TADs, we accessed Hi-C data generated in HMECs[24] through the 3D genome browser (http://promoter.bx.psu.edu/).

**eQTL analysis**. TCGA breast cancer (BRCA) data set was used to test for an association between genotype and mRNA abundance further adjusted for DNA methylation and somatic copy-number profiles. Pre-processed controlled access germline genotype calls (birdseed algorithm) were downloaded from the TCGA data portal. Putative target genes at each of the risk loci were assigned as described above. For SNPs missing from the Affymetrix SNP6 platform, proxy SNPs were identified using phase 3 data from the 1000 Genomes project ($r^2 > 0.8$, distance limit = 500KB). TCGA BRCA mRNA, DNA methylation and DNA copy-number data were downloaded from GDAC (version: 2016_01_28). After excluding data from women of Asian ($N = 37$), African ($N = 159$) or American Indian/Alaska Native ($N = 1$) ethnicity, matched data (including germline genotype data) were available for 547 samples; 415 ER+ samples and 95 ER− samples (ER status was unknown for 37 samples). mRNA data was $\log_2$ transformed for eQTL analysis. Statistical association between mRNA abundance levels and genotype groups (AA, AB, BB) was estimated using multivariate linear regression models with one degree of freedom for genotype groups, adjusted for DNA methylation and copy-number data. For DNA methylation arrays, methylation levels of the probe with strongest inverse correlation (otherwise minimum correlation coefficient) with its target gene's expression were used as representative methylation levels of the target gene. Analyses for ER+ and ER− subsets were performed separately. *P*-values were adjusted for multiple comparisons using the Benjamini–Hochberg method[54]. eQTL analyses in ER+ samples of genes at chromosome 11q13 were further stratified by copy-number gains using the threshold defined in TCGA[25] ($\log_2$ copy number >0.3). The variation in copy number within strata was greatly reduced and the eQTL regression models for these additional stratified analyses were as described above, but adjusted for DNA methylation only.

**Survival analysis**. The Metabric[30] breast cancer cohort (EGA Study ID: EGAS00000000083) was used for DSS analysis. Data were summarised and quantile-normalised from the raw expression files generated by Illumina Bead-Studio (R packages: beadarray v2.4.2 and illuminaHuman v3.db_1.12.2). Raw data files of one Metabric sample were not available at the time of our analysis, and were therefore excluded. The most variable probe was used as a representative for the corresponding gene's mRNA abundance levels. A Cox proportional hazards model was used to estimate pairwise hazard ratios with the lowest expression group treated as baseline. *P*-values for pairwise comparison of survival curves were estimated using Wald tests. The overall test of the Null hypothesis that the expression-derived survival curves show no association with patient outcome was tested with Wald tests (1 degree of freedom, *P*-trend). *P*-values were further adjusted for multiple comparisons using the Benjamini–Hochberg method[54]. Survival analysis was carried out for ER+ and ER− subsets separately.

**Data availability**. All CHi-C data sets generated as part of this analysis are publicly available at (https://www.ebi.ac.uk/ena) under the accession code PRJEB23968. Processed data can be visualised at bit.ly/CHiC-BC/. Publicly available data sets that were accessed for this analysis are detailed in Supplementary Table 3.

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

## Acknowledgements

We thank Prof Georgia Chenevix-Trench for providing Bre80-Q-TERT cells. We thank the Breast Cancer Now Toby Robins Research Centre Bioinformatics Core Facility for Bioinformatics Support. We thank Breast Cancer Now for funding this work as part of Programme Funding to the Breast Cancer Now Toby Robins Research Centre. This work was supported by Cancer Research UK and we acknowledge NIHR funding to the Royal Marsden Hospital Biomedical Research Centre (BRC). F.D. and O.C.L. are additionally supported by the MRC (K006215) and the European Union's Horizon 2020 research and innovation programme under grant agreement No 634570 (FORECEE).

## Author contributions

O.F., J.S.B., N.H.D., N.O., F.D. and S.H. were responsible for the overall design of the study. J.S.B., N.H.D., N.J., V.F., N.S., L.A.M., I.A. and K.F. were responsible for data generation. Bioinformatics analyses were carried out by S.M., S.A., S.W.W., R.C., A.G.R., S.H., O.C.L., F.D., S.H. and O.F. were responsible for the statistical analyses. J.S.B., O.F. and S.H. wrote the manuscript with input from all authors.

## Additional information

**Competing interests:** The authors declare no competing interests.

