## [Peer Review File · Nature Communications]

Reviewers' comments:

Reviewer #1 (Remarks to the Author):

The authors have used Capture HiC to annotate breast cancer risk loci, with the goal of linking noncoding SNPs to target genes. The advantages of this method are that it is genome-wide, scalable to the analysis of multiple regions, and does not require as many sequencing reads as does standard HiC. After identifying SNP-gene interactions, the authors compared the gene lists to somatically mutated genes and also tested whether the SNP:gene links were also identified in eQTL analyses.

The overall goal, as stated on page 10, line 225, is to identify target genes of 63 breast cancer TWAS risk loci and evaluate this comparison to a nearest gene approach. However, it is a bit hard to pull together all the data to actually understand the comparison. You identified 110 putative target genes using CHi-C. Of these, how many were the nearest gene? If you take the list of genes identified by eQTL, how many are the nearest gene? What is the overlap of the two sets? Of the genes that map really far from the SNP, are they in the same TAD or do they cross TAD boundaries?

Clearly the authors have performed a lot of CHi-C experiments in a variety of cell lines. However, it was hard to get an overview of how reproducible the identified loops are. For example, I note that you mention in the methods that you did two biological reps for each cell line, but I could really find any analysis of the reproducibility of the replicates.

In summary, this study represents a great deal of work that could be useful to the breast cancer research community. However, concerns of reproducibility should be addressed.

Other comments:

I'm not sure that Figure 1 is necessary for the main text.

Page 5, lines 100 and 102. You refer to Figure 1c and 1d. These do not exist. Perhaps you mean Figure 2?

Page 6, line 126. You say that you could assign a putative target gene to 33 loci but the Table 2 title says 36 risk loci.

Line 662, 663: You state: "Where TSS for two or more target genes map to a single HindIII fragment, the genes are separated by a comma and the cell lines in which they are targets are given in parenthesis afterwards." This does not seem to match the actual Table; I don't see any cell lines indicated at all.

Table 2: It would be good to indicate what cell line each interaction was identified in and to give the distance between the SNP and the TSS for each interaction.

Figure 2. It is hard to interpret the significance of the overlapping Venn diagrams without knowing how reproducible the protocol is. For example, if you repeated a CHi-C experiment 3 times in one cell line, would you still only see a small percentage of reproducible loops?

It isn't clear to me whether you show any reproducibility in your CHi-C measurements. If you only take loops that are reproducible (either you identify them twice in a single cell line or once in 2 different cell lines), how would that affect the data you present in the graphs in Figure 2? I think that focusing only on reproducible loops is critical.

Page 7, line 160. Why not provide a table of the 9 SNP: gene combinations that you identified via

looping that also have significant eQTL data? It is really hard to pick these out from the supplementary tables.

Clarifications needed for Table 1:

- a. What are the units for median distance? For example, does 1392 refer to 1392 bp or 1392 kb?
- b. Please define "active loci". Does this just mean the number of tested loci (from the original 63) for which you identified loops? If so, I'm not sure that "active" is a correct term- by comparison one might assume that the other loci are inactive but in fact you just might not have captured the loops.
- c. You note the percentage of IPs that are greater than 2MB. It would be also good to include the percentage that are within the same TAD.

Reviewer #2 (Remarks to the Author):

The path from identification of risk factors for a complex disease such as breast cancer to an understanding of the mechanisms that influence, or even cause the disease, is not a simple one. Identifying genes or non-coding RNAs associated with the risk factor loci is the focus of this report. The authors had previously developed capture Hi-C to identify select chromatin interaction peaks for 3 breast cancer risk loci, based on the idea that risk loci likely engage in long-distance interactions with targets of causal variants. Here they test the approach more broadly. They screen 63 established breast cancer loci and reveal that 51 of them have interaction peaks with 33 loci involving 110 putative target genes, both protein coding and non-coding.

To assess support for the identifications, they test associations between (1) level of expression and SNP genotype (eQTL), (2) disease-specific survival (DSS), and (3) relationship to somatic mutations in breast or other cancers. The results show that 22 identified genes are eQTLs, 32 are associated with DSS and 14 correspond to somatic mutations in cancers. Combining all three analyses, 32 genes had support from at least 1 out of the 3 analyses, 6 genes had support from at least 2 of the analyses. The results support the idea that the identified genes may have a causal role in cancer development and the authors argue for their approach as a "first stage" in further validation and mechanistic study of breast cancer causal genes. This proposal should influence thinking in the field. There are places in the manuscript where clarifications could make the conclusions more convincing.

Main issue

Many risk allele capture regions overlap TSS or proximal promoters of target genes. How can authors rule out local effects of these SNPs vs a more distal function? The authors should include a discussion of this point. At the very least, it would be useful to see the distribution of the location of these SNPs in regard to gene proximity.

Other comments

1. Line 67- please make clear the ER status of each cell line in the text.
2. Fig 2C+D- what statistical test is used for significance? It seems dubious that ER- cell lines have significant overlap.
3. Fig3B+D- It is unclear that these figs represent the distal fragments. Please clarify in the graphics.
4. Line 116- This argument is not convincing. Authors show ER+ specific interaction with FGFR2 promoter, but then focus on association between rs2981578 and 2 novel interactions in the same

capture fragment in ZR-75-1 vs T47D. Given that the capture fragments interact with the same distal region in both ER+ and normal cell lines, what is significance of these small gained interactions in the capture fragment? How do these novel interactions affect expression of FGFR2? Can these interactions be recapitulated in-vitro via manipulation of the risk allele?

5. Please highlight location of SNPs on all interaction plots.

6. Lines 169-176- Discussion of interaction of risk allele with FADD. Fig6A- the effect of risk allele on expression looks minimal. What statistical test was used for this analysis. Even if we assume expression is significantly changed, the interactions between FADD and capture fragment are similar in all cell lines. How can the authors explain the effects of distal interactions on expression in the context of risk allele status?

7. Lines 210-217- Discussion of FADD DSS. Once copy-number variant patients are removed from data, how are authors binning low, int, and high expression? Expression in CNV- patients is much lower than CNV+, thus DSS seems to be driven by expression of FADD via copy number gains, not by a subtle shift in expression connected with the risk allele.

Reviewer #3 (Remarks to the Author):

Baxter et al. present a manuscript in which they have performed capture Hi-C on a number of breast cancer risk SNPs. Identifying the downstream targets of a SNP identified in a GWAS is one of the big challenges facing human genetics. The assumption is that risk SNPs should interact with target loci in the 3D genome, therefore ChIC should be a suitable for identifying targets. Interaction partners are identified for risk SNPs and eQTL in large breast cancer compendia is performed to strengthen their observations.

Although the observations could be interesting, two things make it very difficult for me to properly review this paper.

1) Nowhere in the paper is the actual/raw data presented. The only thing that is plotted is the interaction peaks, which are plotted as arcs. This makes it impossible to judge the quality of the data and success of the peak calling.

2) This is especially problematic because the authors use non-standard tools for peak calling. Which is also poorly explained in the methods. The authors should compare their method to standard methods such as CHICAGO and explain why they opted for a different method.

The authors claim:

"On that basis, we would argue that a high-throughput ChI-C analysis can contribute to on-going efforts to functionally annotate GWAS risk loci..."

If they find a risk SNP that supposedly influences gene expression, the authors should introduce this SNP in a cell line that does not harbor this SNP to see if this affects expression. This would be the most powerful way of showing that the association that the authors pick up with ChIC are actually relevant.

The term IP for interaction peak is confusing; please use a different name or acronym.

"Predicated on the assumption that the number of significant IPs at a locus serves as a proxy for activity at that locus,"

Why would this be the case?

Please find below our point by point response to the reviewers' comments:

Reviewer #1 (Remarks to the Author):

The authors have used Capture HiC to annotate breast cancer risk loci, with the goal of linking noncoding SNPs to target genes. The advantages of this method are that it is genome-wide, scalable to the analysis of multiple regions, and does not require as many sequencing reads as does standard HiC. After identifying SNP-gene interactions, the authors compared the gene lists to somatically mutated genes and also tested whether the SNP:gene links were also identified in eQTL analyses.

The overall goal, as stated on page 10, line 225, is to identify target genes of 63 breast cancer GWAS risk loci and evaluate this comparison to a nearest gene approach. However, it is a bit hard to pull together all the data to actually understand the comparison.

(1) You identified 110 putative target genes using CHi-C. Of these, how many were the nearest gene?

Response: In the original text, we tried to present this comparison i.e. "CHi-C target genes" versus "nearest genes" in Table 2 (final column "Agrees") and in the text using three groups:

Nearest gene is the sole CHi-C target gene (\checkmark ; N=9)

Nearest gene is one of several CHi-C target genes ($\checkmark+$; N=15)

Nearest gene is not a CHi-C target gene (X; N=12)

In response to the reviewer's comment (and taking account of this reviewer's comment 12, below) we have revised Table 2 so that it is entirely focussed on the comparison of "CHi-C target genes" versus "nearest genes" (and no longer includes information about eQTLs as well). To further clarify, we have re-named the third column of Table 2 "CHi-C target genes". The fourth column still shows the "nearest gene" but now when the nearest gene is included in the list of CHi-C target genes it is in bold. The final column still summarises this comparison as \checkmark , $\checkmark+$ or X.

We have also amended the text (lines 156 to 162) so that it is clear that of the 36 loci at which we can compare CHi-C target genes and nearest genes there are 24 loci (and hence, 24 genes) at which the nearest gene is also a CHi-C target gene.

This section of text now reads:

Of the 36 loci at which we were able to assign at least one target gene directly (N=33) or indirectly (N=3) there were 24 at which the nearest gene was either the only CHi-C target gene (N=9; Table 2) or one of several CHi-C target genes (N=15; Table 2). There were, however, 12 loci at which our data implicated genes other than the nearest gene; these loci included 13q13.1-rs11571833 (CHi-C gene: *PDSB5*, nearest gene: *BRCA2*), 14q24.1-rs2588809 and rs999737 (CHi-C gene: *ZFP36L1*, nearest gene: *RAD51B*) and 16q12.2-rs17817449 and rs11075995 (CHi-C genes: *CRNDE*, *IRX5*, *IRX3*, *LOC100996*, nearest gene: *FTO*).

Please also see our response to this reviewer's comment 7, below regarding the 36 loci.

(2) If you take the list of genes identified by eQTL, how many are the nearest gene? What is the overlap of the two sets?

Response: In response to this comment (and this reviewer's comment 12, below) we have provided a new table summarising the statistically significant eQTLs (FDR corrected $P < 0.1$). Of the nine statistically significant eQTLs with CHI-C target genes, there were three at which the CHI-C target gene was also the nearest gene and six at which the CHI-C target gene was not the nearest gene. These are shown in Table 3, in bold.

With regard to overlap between the two sets, we did not originally run a separate eQTL analysis of "nearest genes" – and it would be difficult to compare a separate "nearest genes" analysis with our CHI-C targets analysis as the FDR correction of P-values would differ between analyses. To answer this comment, however, we have re-run our eQTL analysis with the addition of "nearest genes" that were not selected as CHI-C target genes.

These are 2q35-rs13387042 **TNP1**, 2q35-rs16857609 **TNS1**, 3p26.1-rs6762644 **ITPR1**, 6q25-rs12662670, rs2046210 **CCDC170**, 8q24.21-rs13281615 **POU5F1B**, 8q24.21-rs11780156 **PVT1**, 10p12.31-rs7072776, rs11814448 **DNAJC1**, 14q24.1-rs2588809 **RAD51B**, 14q24.1-rs999737 **RAD51B**, 16q12.2-rs17817449, rs11075995 **FTO**, 19p13.1-rs8170, rs2363956 **BABAM1**, **ANKLE1**, 19p13.11-rs4808801 **ELL**, 22q13.1-rs6001930 **MLK1**.

TNP1 is not expressed at detectable levels in breast tissue and was therefore excluded. Of the other additional 18 SNP and "nearest gene" combinations that we tested, two (rs4808801_ **ELL** and rs8170_rs34084277_ **ANKLE1**) were eQTLs in an all cancers analysis and one (rs4808801_ **ELL**) was an eQTL in an analysis of ER+ cancers (FDR corrected $P < 0.1$). We have added this information as a footnote to the new eQTL table (Table 3) and added a sentence to the text with this additional information (line 177 to 181).

This section of text now reads:

Including all nearest genes (regardless of whether they were also a CHI-C target gene) in our eQTL analysis we found two additional SNP-gene combinations that were not captured by our CHI-C analysis; rs4808801 was associated with levels of expression of **ELL** in all cancers and ER+ cancers and rs8170 was associated with levels of expression of **ANKLE** in all cancers (FDR adjusted $P < 0.1$).

(3) Of the genes that map really far from the SNP, are they in the same TAD or do they cross TAD boundaries?

Response: We cannot map TADs using our own CHI-C data – it requires Hi-C data. While considering how best to respond to this comment (and this reviewer's comments re clarifications to Table 1 (c)) we therefore examined TAD boundaries at several of our risk loci using publicly available data and the corresponding algorithms that the authors who generated these datasets used in their analyses. Consistent with the recent paper from Dali and Blanchette (Nucleic Acids Research (2017) 45 2,994 – 3,005) in which they compared seven of the currently available TAD prediction tools we found that the results of these analyses varied considerably depending on the algorithm used. Given this variation, we are reluctant to try to classify our interaction peaks according to TADs (as per the reviewer's comment regarding clarifications to Table 1 (c)) but we accept that some reference to TADs (particularly with respect to the long-range interaction peaks) is important.

Accordingly, we accessed Hi-C data generated in human mammary epithelial cells (the closest approximation to our cell lines that is available) by Rao and colleagues (Cell 159 1665-1680) and

generated Supplementary Figures (3a – d) showing our interaction peaks aligned with TADs for four risk loci at which these long-range interaction peaks (>2Mb) with gene targets map. These are 3p26.1-rs6762644 (*CAV3*; *LINC00312*; *LMCD1* *c3orf32*; *RAD18*; *SETD5*), 4q24-rs9790517 (*CENPE*), 8q24.21-rs13281615 (*CCDC26*), 11q13.1-rs3903072 (*CCND1*, *FADD*). From these figures, the reader can now see that these loci form interaction peaks between (i) captured fragments and a target gene that maps within the same TAD but also (ii) captured fragments and a target gene that map to different TADs. We have added text (line 145 -151) referring to this analysis and these supplementary figures.

This additional section of text reads:

For the interaction peaks with the longest-range gene targets (3p26.1-rs6762644 (*CAV3*, *LINC00312*, *LMCD1*, *c3orf32*, *RAD18*, *SETD5*), 4q24-rs9790517 (*CENPE*), 8q24.21-rs13281615 (*CCDC26*), 11q13.1-rs3903072 (*CCND1*, *FADD*)) we aligned our data with topologically associated domains (TADs) generated in Human Mammary Epithelial Cells (HMEC)²². At each of these loci we observed interaction peaks between captured fragments and target gene(s) mapping within the same TAD but also, less frequently, with target gene(s) mapping to a different TAD (Supplementary Figure 3).

(4) Clearly the authors have performed a lot of ChI-C experiments in a variety of cell lines. However, it was hard to get an overview of how reproducible the identified loops are. For example, I note that you mention in the methods that you did two biological reps for each cell line, but I could really find any analysis of the reproducibility of the replicates.

In summary, this study represents a great deal of work that could be useful to the breast cancer research community. However, concerns of reproducibility should be addressed.

Response: Please see our response to comment (11), below regarding the reproducibility of biological replicates.

Other comments:

(5) I'm not sure that Figure 1 is necessary for the main text.

Response: We have moved this figure to supplementary.

(6) Page 5, lines 100 and 102. You refer to Figure 1c and 1d. These do not exist. Perhaps you mean Figure 2?

Response: Thank you – we have corrected this – although ironically these are now Fig 1c and 1d as we have moved Fig 1 to Supplementary.

(7) Page 6, line 126. You say that you could assign a putative target gene to 33 loci but the Table 2 title says 36 risk loci.

Response: We apologise for this confusion. We were able to assign putative target genes to 33 risk loci directly. We were also able to assign putative target genes to three additional risk loci on the basis of interaction peaks with adjacent risk loci (8q21.11-rs6472903 with *HNF4G/PEX2*, 9q31.2-rs10759243 with *KLF4* and 14q24.1-rs999737 with *ZFP36L1*). We have clarified this in the text (line 156 - 158) and in the Table 2 title.

(8) Line 662, 663: You state: "Where TSS for two or more target genes map to a single HindIII

fragment, the genes are separated by a comma and the cell lines in which they are targets are given in parenthesis afterwards.” This does not seem to match the actual Table; I don’t see any cell lines indicated at all.

Response: We apologise, this footnote referred to a previous version of the table. We removed this information as the table became over complicated – but please see response to (9) below.

(9) Table 2: It would be good to indicate what cell line each interaction was identified in and to give the distance between the SNP and the TSS for each interaction.

Response: We have added two new Supplementary Tables - Supplementary Table 4 which shows the cell lines in which each interaction peak was identified and Supplementary Table 5 which gives the distance between the reported risk SNP and the TSS for each interaction.

(10) Figure 2. It is hard to interpret the significance of the overlapping Venn diagrams without knowing how reproducible the protocol is. For example, if you repeated a CHi-C experiment 3 times in one cell line, would you still only see a small percentage of reproducible loops?

And, reviewer #2, “other comments” (2)

Response: I’m afraid we can’t answer this directly, as we haven’t repeated the same CHi-C experiment 3 times. However, we have now provided heatmaps illustrating the reproducibility of the raw data between biological duplicates (Supplementary Figures 6 and 7 and added Supplementary Table 3 detailing the number and proportion of interaction peaks that we report which are reproduced exactly in multiple cell lines (see response to comments 4 and 11 below).

This reviewer’s comment also raises an issue regarding the clarity of this section. The Venn diagrams presented in Figure 2 (now Figure 1) were originally based on **interacting fragments** i.e. the fragments at each end of an interaction peak, not the interaction peaks (loops). Given the emphasis on reproducibility of interaction peaks in the reviewers’ comments, however, we have revised this analysis so that it is now based on interaction peaks. The results we now report for interaction peaks (loops) provide strong evidence for a subset of interaction peaks that are remarkably reproducible between all four breast cancer cell lines ($N = 62$, $P = 0.0003$) and between all five breast specific cell lines ($N = 53$, $P < 0.0001$). We also demonstrate that there is an excess of interaction peaks in lymphoblastoid cell lines that are not shared with the breast specific cell lines suggesting a degree of cell type specificity. There is some evidence that the ER+ breast cell lines are more similar to each other than they are to the ER- breast cancer cell lines – but the evidence of specificity by receptor status is less strong (lines 108 – 114 and Figures 1c – 1e).

This section of text now reads

We found a statistically significant excess of interaction peaks that were common to all four breast cancer cell lines ($N = 62$, $P = 0.0003$, Figure 1c) and all five breast cell lines ($N = 53$, $P < 0.0001$ Figure 1d). We also found an excess of interaction peaks that were exclusive to the lymphoblastoid cell line ($N = 304$, $P < 0.0001$) suggesting that at least a subset of interaction peaks show cell type specificity. Comparing the cell lines according to receptor type, the interaction peaks were marginally more similar within the two ER+ cell lines (Jaccard similarity coefficient = 0.18) and the two ER- cell lines (Jaccard similarity coefficient = 0.13) than between them (Figure 1e).

Comments (4) and (11)

(4) Clearly the authors have performed a lot of CHi-C experiments in a variety of cell lines. However, it was hard to get an overview of how reproducible the identified loops are. For example, I note that you mention in the methods that you did two biological reps for each cell line, but I could really find any analysis of the reproducibility of the replicates.

In summary, this study represents a great deal of work that could be useful to the breast cancer research community. However, concerns of reproducibility should be addressed.

And

(11) It isn't clear to me whether you show any reproducibility in your CHi-C measurements. If you only take loops that are reproducible (either you identify them twice in a single cell line or once in 2 different cell lines), how would that affect the data you present in the graphs in Figure 2? I think that focusing only on reproducible loops is critical.

Response: To maximise the power of our analysis we combine data from the two biological replicates from a single cell line before calling our interaction peaks. In effect, therefore, an interaction peak will only be called if there is support for this combination of interacting fragments in both biological replicates. This means that we cannot formally test the reproducibility of called interaction peaks within a cell line. We can, however, test the reproducibility of (raw) di-tags at each pairwise combination of Hind III fragments in the two replicates from each cell line.

In response to comment (4), therefore, we have analysed the correlation between replicates, stratifying on the genomic distance between the two interacting fragments and according to the two types of analysis (ligations between a captured fragment and (i) another captured fragment in *cis* or (ii) a non-captured fragment in *cis*, mapping within five Mb).

Predictably, the correlation between replicates is strongest when the two fragments are closer and both fragments are captured. When the two fragments map within 500 kb of each other Spearman's ρ ranges from 0.78 to 0.92 (both fragments captured) and 0.58 to 0.77 (just one fragment captured). For fragments separated by 500 kb to 1Mb, Spearman's ρ ranges from 0.53 to 0.77 (both fragments captured) and 0.33 to 0.59 (just one fragment captured). For fragments that are separated by more than 1Mb, i.e. where most of the (raw) di-tags represent noise, there is no correlation (Spearman's ρ ranges from -0.38 to + 0.37). We have added this analysis to the Methods (lines 424 - 434) and provided Supplementary Figure 6 (both fragments captured) and 7 (one fragment captured).

In response to comment (11): As explained above, our method of analysis means that we cannot use the reproducibility of interaction peaks within a cell line explicitly as a metric for selecting which interaction peaks (or loops) to report. It is, however, implicit in the analysis. We further minimise our type I errors by using a strict threshold for statistical significance (FDR corrected $P < 0.01$) and we only "call" a putative CHi-C target gene if the TSS of the gene is involved in interaction peaks in at least two cell lines (albeit not always with identical partner fragments).

We accept that providing the reader with a set of loops that are identical in two separate cell lines would be useful but we think that not even reporting the other interaction peaks would be restrictive. We will illustrate this using the putative Hi-C target gene *FADD* as an example. The TSS of

FADD is involved in interaction peaks in all five breast cell lines, comprising nine individual loops. Five of these loops occur in just one cell line, two occur in two cell lines, one occurs in three cell lines and one occurs in four cell lines; several of the captured interacting fragments are consecutive (rather than identical). For the majority of the interaction peaks the captured fragment maps to 11q13.1-rs3903072 and the interaction peaks span approximately 550 kb. There are, in addition, three interaction peaks, all in ZR-75-1, for which the captured interacting fragment maps to 11q13.3-rs554219 and for which the interaction peak spans 4.5 Mb. There are also 10 interaction peaks between 11q13.1-rs3903072 and 11q13.3-rs554219 spanning ~4Mb – none of these is identical between cell lines but they cluster to a subset of fragments within the 11q13.3-rs3903072 capture region. We think that presenting these data in full is useful; it could help to inform follow up studies characterising these two risk loci and evaluating *FADD* as a putative breast cancer gene.

To respond to comment (11), however, we have generated a data set comprising just the subset of loops that are identical in at least two separate cell lines. These are provided as a supplementary dataset in a format that can be uploaded onto the WashU website.

(12) Page 7, line 160. Why not provide a table of the 9 SNP: gene combinations that you identified via looping that also have significant eQTL data? It is really hard to pick these out from the supplementary tables.

Response: We have done this, this is now Table 3

Clarifications needed for Table 1:

a. What are the units for median distance? For example, does 1392 refer to 1392 bp or 1392 kb?

Response – we have added the units (kb)

b. Please define “active loci”. Does this just mean the number of tested loci (from the original 63) for which you identified loops? If so, I’m not sure that “active” is a correct term- by comparison one might assume that the other loci are inactive but in fact you just might not have captured the loops.

Response – we accept this comment and agree that “active” is not the most appropriate term. We have altered our terminology to “informative” and amended the text and tables accordingly.

c. You note the percentage of IPs that are greater than 2MB. It would be also good to include the percentage that are within the same TAD.

Please see response to comment (3) above.

Reviewer #2 (Remarks to the Author):

The path from identification of risk factors for a complex disease such as breast cancer to an understanding of the mechanisms that influence, or even cause the disease, is not a simple one. Identifying genes or non-coding RNAs associated with the risk factor loci is the focus of this report. The authors had previously developed capture Hi-C to identify select chromatin interaction peaks for

3 breast cancer risk loci, based on the idea that risk loci likely engage in long-distance interactions with targets of causal variants. Here they test the approach more broadly. They screen 63 established breast cancer loci and reveal that 51 of them have interaction peaks with 33 loci involving 110 putative target genes, both protein coding and non-coding.

To assess support for the identifications, they test associations between (1) level of expression and SNP genotype (eQTL), (2) disease-specific survival (DSS), and (3) relationship to somatic mutations in breast or other cancers. The results show that 22 identified genes are eQTLs, 32 are associated with DSS and 14 correspond to somatic mutations in cancers. Combining all three analyses, 32 genes had support from at least 1 out of the 3 analyses, 6 genes had support from at least 2 of the analyses. The results support the idea that the identified genes may have a causal role in cancer development and the authors argue for their approach as a “first stage” in further validation and mechanistic study of breast cancer causal genes. This proposal should influence thinking in the field. There are places in the manuscript where clarifications could make the conclusions more convincing.

Main issue

Many risk allele capture regions overlap TSS or proximal promoters of target genes. How can authors rule out local effects of these SNPs vs a more distal function? The authors should include a discussion of this point. At the very least, it would be useful to see the distribution of the location of these SNPs in regard to gene proximity.

Response: In response to this comment (and Reviewer 1, comment 9) we have added Supplementary Table 5 with the distances between the SNPs and the putative CHI-C target genes. We have also added a discussion of this point as requested (lines 291 - 295):

This new section reads

While the presence of these long range interactions may inform future follow up studies, they do not exclude effects that are more local to the risk loci. In their functional annotation of the human genome, the ENCODE consortium estimated that the average number of TSSs that interact with any given distal element is 2.5³⁵ and regulatory variants that map to such elements may influence absolute or relative levels of expression of multiple genes.

Other comments

1. Line 67- please make clear the ER status of each cell line in the text.

Response: We have added this information.

2. Fig 2C+D- what statistical test is used for significance? It seems dubious that ER- cell lines have significant overlap.

Response: The test we used was a permutation test. These Venn diagrams were also raised by reviewer #1, comment 10 and we have revised this analysis. In the revised analysis we still use permutation but the unit of analysis is now “loops” rather than interacting fragments and we have added Figure 1e showing the Jaccard similarity coefficients for ER+ and ER- breast cancer cell lines.

The ER- cell lines do have overlap (Jaccard similarity coefficient = 0.13) – but the reviewer is correct in observing that this is not strong. This is now stated in the text and the figures have been revised.

This section of text (line 108 – 114) now states:

We found a statistically significant excess of interaction peaks that were common to all four breast cancer cell lines (N = 62, P = 0.0003, Figure 1c) and all five breast cell lines (N = 53, P < 0.0001 Figure 1d). We also found an excess of interaction peaks that were exclusive to the lymphoblastoid cell line (N = 304, P < 0.0001) suggesting that at least a subset of interaction peaks show cell type specificity. Comparing the cell lines according to receptor type, the interaction peaks were marginally more similar within the two ER+ cell lines (Jaccard similarity coefficient = 0.18) and the two ER- cell lines (Jaccard similarity coefficient = 0.13) than between them (Figure 1e).

3. Fig3B+D- It is unclear that these figs represent the distal fragments. Please clarify in the graphics.

Response we have added “enlarged in” to the graphics.

4. Line 116- This argument is not convincing. Authors show ER+ specific interaction with FGFR2 promoter, but then focus on association between rs2981578 and 2 novel interactions in the same capture fragment in ZR-75-1 vs T47D. Given that the capture fragments interact with the same distal region in both ER+ and normal cell lines, what is significance of these small gained interactions in the capture fragment? How do these novel interactions affect expression of FGFR2? Can these interactions be recapitulated in-vitro via manipulation of the risk allele?

Response: We agree, this is very speculative and we are not able to provide *in vitro* data to support this argument. We have removed this section and the associated Supplementary Figure 2 from the paper.

5. Please highlight location of SNPs on all interaction plots.

Response: We have added the location of the GWAS SNPs to the interaction plots but we should probably emphasise that these are just tag SNPs and there is no implication that they are causal (please see response to reviewer #3, comment 3).

6. Lines 169-176- Discussion of interaction of risk allele with FADD. Fig 6A - the effect of risk allele on expression looks minimal. What statistical test was used for this analysis.

Even if we assume expression is significantly changed, the interactions between FADD and capture fragment are similar in all cell lines. How can the authors explain the effects of distal interactions on expression in the context of risk allele status?

Response: The statistical test was a t-test of the null hypothesis that the partial regression coefficient (β_{genotype}), from a multiple regression model that includes genotype, methylation and copy number, differs from zero. Unadjusted P values (0.04 for all cancers and 0.01 for ER+ cancers) are

presented and the FDR corrected P values (0.28 for all cancers, 0.17 for ER+ cancers) are also presented.

We agree – the effect is minimal – but there are two issues that need to be taken into account

(i) the P values are from a multiple regression model in which both “methylation” and “copy number” will explain a proportion of the variance (possibly a statistically significant proportion).

(ii) these are common low penetrance variants which (individually) have small effects on risk. For example, the variant allele of rs3903072 is associated with an approximately 5% (95% Ci 4% to 7%) reduction in breast cancer risk.

Physical interactions between – for example – an enhancer and a promoter are dynamic. The interaction peaks that we observe represent a time-averaged picture across a large number of cells. Whilst in an extreme situation, the one allele of a causal variant might completely prevent a particular physical interaction from occurring, it is much more likely that it would be one factor (among many) that would alter the frequency with which we observe that interaction. We believe the reviewer has raised an important issue regarding the effect sizes of these risk loci and we have added a sentence to the discussion (line 309 - 314) that addresses this point and is pertinent to this reviewer’s comment 7, below.

This additional text reads

The variants detected by GWAS are common variants with small effects (ORs are typically < 1.2) and any individual risk SNP will usually only explain a small proportion of variance in levels of expression of a target gene. For example, the association between 11q13.1-rs3903072 and *FADD* is weak in all ER+ cancers (nominal $P = 0.01$); excluding ER+ cancers with copy number gains reduces the variance in levels of expression of *FADD* and increases the proportion of variance explained by rs3903072 (nominal $P = 0.004$).

7. Lines 210-217- Discussion of FADD DSS. Once copy-number variant patients are removed from data, how are authors binning low, int, and high expression? Expression in CNV- patients is much lower than CNV+, thus DSS seems to be driven by expression of FADD via copy number gains, not by a subtle shift in expression connected with the risk allele.

Response: After excluding the copy-number variant patients from our eQTL analyses, the variation in \log_2 copy number is markedly reduced and we no longer bin/adjust the remaining samples for copy number variation. The section of the methods that referred to this stratified analysis was not clear enough in our original description so we have reworded this section (lines 523 -527) so that it now reads:

eQTL analyses, in ER+ samples, of genes at chromosome 11q13 were further stratified by copy-number gains using the threshold defined in TCGA²³ (\log_2 copy number > 0.3). The variation in copy number within strata was greatly reduced and the eQTL regression models for these additional stratified analyses were as described above, but adjusted for DNA methylation only.

Reviewer #3 (Remarks to the Author):

Baxter et al. present a manuscript in which they have performed capture Hi-C on a number of breast

cancer risk SNPs. Identifying the downstream targets of a SNP identified in a GWAS is one of the big challenges facing human genetics. The assumption is that risk SNPs should interact with target loci in the 3D genome, therefore CHi-C should be a suitable method for identifying targets. Interaction partners are identified for risk SNPs and eQTL in large breast cancer compendia is performed to strengthen their observations.

Although the observations could be interesting, two things make it very difficult for me to properly review this paper.

1) Nowhere in the paper is the actual/raw data presented. The only thing that is plotted is the interaction peaks, which are plotted as arcs. This makes it impossible to judge the quality of the data and success of the peak calling.

Response: We thank the reviewer for this suggestion - it has forced us to think about how to present our raw data aligned to the results of our analysis. There are difficulties showing this type of CHi-C data (in which we capture a series of consecutive HindIII fragments) as opposed to promoter CHi-C data (where single promoter fragments are captured). Our new Supplementary Figures 8 and 9 show raw data for the 14 captured fragments that map to the 10q26.13 risk locus (i.e. what is now Figure 2a in the main text) and the 13 captured fragments that comprise the 11p15.5 risk locus (i.e. what is now Fig 3a).

Raw data for the di-tags formed by each consecutive captured fragment are shown as separate rows in the new Supplementary Figures. We have aligned these to the loops (that represent statistically significant interaction peaks) from the respective (main) Figures. By doing so we are able to illustrate that the ends of the loops co-localise with peaks in the raw data.

However, it is important to point out that these peaks vary between rows i.e. the called interaction peaks are specific to particular combinations of captured and non-captured HindIII fragments. For example, at 10q26.13 the majority of interaction peaks are targeting three uncaptured consecutive HindIII fragments mapping at 122,701,138-122,708,722 bps (shown in detail in Fig 3b); interaction peaks are clearly called for captured fragments T-47D 8-10 (ie rows T47D 8-10 in Supplementary Figure 8).

However, the raw data can – to some extent - be misleading. While it is clear that the raw peaks and called loops align, the statistical significance of the raw peaks depends on the “interactability” of the captured fragment (represented in each row). T-47D fragments 12 – 14 show low “interactability” (compare the reads immediately adjacent to the capture region in these rows with the rows above) whereas T-47D fragments 2 & 3 have very high levels of “interactability”. Accordingly, the interaction peaks are significant in rows 12-14 but not 2 & 3. There are no statistically significant interaction peaks at this locus in MDA-MB-231.

For 11p15.5 the predominant interaction peaks target uncaptured HindIII fragments mapping to 2169843 – 2173106 bps (colocalising with the *IGF2* TSS) and 1647384 – 1659371 bps (colocalising with the *KRTAP5-5* TSS) in MDA-MB-231. The peaks are specifically called for captured fragments MDA 2 – 5 and 11 – 13 (*IGF2*) and MDA 12 – 13 (*KRTAP5-5*). There are no significant interaction peaks targeting *IGF2* and just a single interaction peak targeting *KRTAP5-5* in T-47D.

2) This is especially problematic because the authors use non-standard tools for peak calling. Which

is also poorly explained in the methods. The authors should compare their method to standard methods such as CHICAGO and explain why they opted for a different method.

Response: We published the first GWAS CHi-C paper as a method paper (Dryden N et al Genome Res 24, 1854-68) in 2014. Here we have applied the same method of analysis that we developed in that study, basing it on the 5C analysis method used by Sanyal and colleagues (Nature 489 109 - 113). Since then, there have been 16 CHi-C papers published; they comprise two different experimental designs (GWAS CHi-C and promoter CHi-C) and use four different analysis methods (our method, one based on the iterative correction method for Hi-C data published by Imakaev and colleagues (Imakaev M et al 2012, Nat Methods 9 999-1003), GOTHIC (Mifsud B et al 2017, PLoS One 12(4) e0174744) and CHICAGO); other groups have published using our method (Martin P et al 2015, Nat Commun 30;6:10069).

Hi-C analysis methods were recently compared in a Nature Methods paper (Forcato et al 2017, Nature Methods 14 679 – 685) which concluded that there is no single method that outperforms all other methods. However, the CHi-C analysis methods have not been formally compared and it is premature to call one method standard and another non-standard. We accept that a formal comparison of methods is needed, but the focus of our paper is identifying putative target genes for breast cancer GWAS risk loci, and carrying out a full comparison of CHi-C analysis methods is beyond its scope.

Both our approach and CHICAGO are based on a negative binomial regression and the differences are mainly in the error model, which would tend to affect power more than the false positive rate. We would also like to note that in unpublished work, Dr Chris Wallace and colleagues [1] have compared our approach to CHICAGO in a bidirectional design in which the uncaptured targets interacting with capture regions are then themselves captured in a reverse experiment. Our approach was found to be more reliable than CHICAGO at detecting the same interacting pairs in both experiments – presumably, the true positives. Taking these considerations together we believe that our method is sound and not demonstrably inferior to any existing alternative.

[1] Eijsbouts C (2016) Consistent Detection of Chromatin Contacts. MPhil in Computational Biology dissertation, University of Cambridge (dissertation available upon request)

To help the reader to follow the methodology, we have added an overview of the analysis to the beginning of the Methods section (lines 455 - 459) and added that the R scripts are available on request.

(3) The authors claim:

“On that basis, we would argue that a high-throughput CHi-C analysis can contribute to on-going efforts to functionally annotate GWAS risk loci...”

If they find a risk SNP that supposedly influences gene expression, the authors should introduce this SNP in a cell line that does not harbor this SNP to see if this affects expression. This would be the most powerful way of showing that the association that the authors pick up with CHi-C are actually relevant.

Response: We accept that the gold standard for calling a SNP functional would be generating isogenic cell lines with all three combinations of alleles (AA, Aa and aa) and testing for an association with gene expression. However, the SNPs that informed this analysis are just GWAS hits, i.e. tag SNPs that capture common variation at a given locus. This is why we have captured linkage disequilibrium blocks rather than individual HindIII fragments that harbour the GWAS SNPs.

While one or two GWAS SNPs might also be causal, the majority will be associated with outcome because they are in linkage disequilibrium with a true causal variant. Similarly, linkage will drive the eQTL associations. Carrying out genome editing of GWAS SNPs (rather than putative causal variants) is unlikely to shed light on mechanism. To generate lists of likely causal variants, fine-scale mapping of these loci using a custom chip (OncoArray) in approximately 60,000 cases and 60,000 controls is currently underway as part of a large international consortium (the Breast Cancer Association Consortium). These data should be available in the next 12 – 18 months.

(4) The term IP for interaction peak is confusing; please use a different name or acronym.

Response: We have removed the term IP and replaced it with “interaction peak”.

(5) “Predicated on the assumption that the number of significant IPs at a locus serves as a proxy for activity at that locus,”

Why would this be the case?

Response: In response to this comment and with reference to reviewer 1, comments on Table 1 (b) we accept that the concept of “active” loci is misleading. We have removed this text so that this section now reads (line 88 onwards)

“We first tested for differences in the median number of interaction peaks....”

Finally, during the revision of this manuscript we realised that our original description of Figure 6e (line 217 in the original manuscript) was incorrect. In fact, it is the common allele of rs1550623 that is associated with lower levels of CDCA7 expression (compare heterozygotes with common homozygotes, ignoring the two rare homozygotes). We have revised this section of the results (now lines 239 – 242) so that it states:

Similarly, for *CDCA7* the associations are inconsistent. The risk allele of rs1550623 is the common allele²⁴; the common allele is associated with lower levels of *CDCA7* expression (Figure 5e) but lower levels of expression of *CDCA7* are associated with a better prognosis (Figure 5f).

Reviewers' comments:

Reviewer #1 (Remarks to the Author):

The authors have addressed most of the questions and concerns. I do have one remaining question that the authors could address in their final version.

In Table 1. Is it possible that the higher percentage of loops > 2 Mb in T47D and ZR-75-1 cells are due to a higher frequency of chromosomal rearrangements or other abnormalities (large deletions between the loops) in these cells?

Reviewer #2 (Remarks to the Author):

Authors have provided thoughtful and thorough responses to comments of referees.

Reviewer #3 (Remarks to the Author):

The authors have answered my questions.

It would be great if the authors could add the orientation of the CTCF binding sites to identified interactions (e.g. Fig. 2b,d). It would be interesting to stratify the observed interactions based on CTCF-CTCF interaction (with orientation) seeing as both interaction shown have CTCF binding.

Reviewer #1 (Remarks to the Author):

The authors have addressed most of the questions and concerns. I do have one remaining question that the authors could address in their final version.

In Table 1. Is it possible that the higher percentage of loops > 2 Mb in T47D and ZR-75-1 cells are due to a higher frequency of chromosomal rearrangements or other abnormalities (large deletions between the loops) in these cells?

Response: This is an important point, and we thank the reviewer for raising it. The breast cancer cell lines (T-47D, ZR-75-1, BT-20 and MDA-MB-231) will all have chromosomal rearrangements, gains and losses that are absent from the "normal" cell lines (Bre80 and GM06990). Based on the extensive analysis of genetic aberrations in both tumours and cell lines carried out by Neve and colleagues (Cancer Cell 2006 10(6) 515-27) there is no evidence that luminal (ER+) breast cancer cell lines carry more genome aberrations than basal (ER-) breast cancer cell lines. However, we cannot exclude the possibility that rearrangements, gains and losses occur preferentially at ER+ risk loci in ER+ cell lines and that this contributes to the higher proportion of long range interaction peaks we observe in the ER+ breast cancer cell lines.

We have added this information to the revised text (page 5, lines 99-104) and added the paper by Neve and colleagues to the references.

Reviewer #2 (Remarks to the Author):

Authors have provided thoughtful and thorough responses to comments of referees.

Response: Thank you.

Reviewer #3 (Remarks to the Author):

The authors have answered my questions.

It would be great if the authors could add the orientation of the CTCF binding sites to identified interactions (e.g. Fig. 2b,d). It would be interesting to stratify the observed interactions based on CTCF-CTCF interaction (with orientation) seeing as both interaction shown have CTCF binding.

Response: We have added the direction of the CTCF binding sites (based on CTCF ChIP-seq data generated in T-47D cells) to Figures 2b and 2d. We have revised the figure legends to explain this and we have added the information that for these two examples the orientation of the CTCF binding sites is towards the captured locus (page 6, line 127-128).

However, ChI-C data (as opposed to Hi-C data) only samples a subset of all possible CTCF-CTCF interactions (ie we only see those where the interaction involves at least one bait fragment that maps to a risk locus). For example, at the 10q26.13 locus we see the two CTCF binding sites that map

to chr10:122,701,000-122,709,000 (Figure 2b) because they form interaction peaks with captured fragments mapping to the risk locus at chr10:123,329,667 - 123,379,947. However, we don't see four other CTCF binding sites (chr10:122,829,891 - 122,830,131, chr10:122,848,063 - 122,848,303, chr10:122,928,389 - 122,928,629 and chr10:123,152,195- 123,152,435) that map in between the distal CTCF binding sites in Figure 2b and the capture region. As the CTCF-CTCF interactions we observe represent a (non-random) sample of all possible CTCF-CTCF interactions, it is not clear what this subset represents and we would prefer not to stratify our interaction peaks on this basis.